# Learning Context-Conditioned Predicate Semantics via Prototype Feedback

**NamGyu Jung** [1]    **Chang Choi** [1]

## Abstract

In scene graph generation, a central challenge is modeling polysemous predicates whose meanings shift across contexts. Prior approaches address this issue by decomposing predicates into multiple static prototypes or retrieving semantically similar exemplars. However, these strategies keep predicate representations static and cannot reorganize semantics to reflect image-specific evidence, leading to systematic confusions in ambiguous contexts. We propose **AlignG**, which learns context-conditioned predicate semantics via prototype feedback. AlignG infers context-conditioned predicate semantics from the relation candidates within each image and feeds the adapted semantics back to recalibrate relation representations. The learning objective anchors this adaptation to global semantic centers, preventing semantic drift while still allowing selective reorganization when the scene provides consistent relational cues. Experiments on VG-150 and GQA-200 show consistent improvements over state-of-the-art baselines, with F@100 improvements of +1.4 on VG-150 and +2.7 on GQA-200 under SGDet. We further visualize per-image prototype similarity shifts and observe coherent context-dependent reorganization where prototypes selectively merge or separate predicates according to scene evidence. The code is available at https://github.com/Namgyu97/AlignG-SGG.pytorch.

## 1. Introduction

Scene Graph Generation (SGG) has become a central task for structured scene understanding, where an image is represented as a graph of objects and their semantic relationships (Xu et al., 2017). By capturing interactions such as spatial, functional, and compositional relations, SGG provides structured representations that support downstream reasoning tasks including visual question answering, image retrieval, and robotic planning (Zellers et al., 2018). In real-world imagery, however, predicates are often polysemous and the same label such as "on" may indicate spatial contact in one case and functional usage in another. This semantic variability, exacerbated by the long-tail distribution of relations, poses a fundamental obstacle to both accurate prediction and contextual interpretation (Krishna et al., 2017; Tang et al., 2020; Yu et al., 2021).

To address semantic variability, prototype-based methods introduce fixed anchors to structure the predicate space. Early models such as PE-Net (Zheng et al., 2023) adopt a single prototype per predicate, while later work either decomposes predicates into multiple prototypes (Chen et al., 2024) or integrates structured knowledge (Zareian et al., 2020; Zhang et al., 2024). For example, the predicate "riding" can be split into variants such as standing, crouching, or sitting, each tied to a distinct posture within the same relation (Zheng et al., 2023; Jeon et al., 2024). Yet, as illustrated in Figure 1, these decompositions remain insufficient when meaning depends on the surrounding environment. A skier moving downhill may indeed be riding the skis, whereas individuals who simply stand on skis without motion are not, and fixed prototypes cannot adjust to such contextual differences.

In response to these limitations, recent studies have explored auxiliary signals and architectural innovations. Retrieval-based methods search for semantically similar exemplar cases, improving recognition of rare or long-tail relations (Yoon et al., 2025). Large language models (LLMs) provide external commonsense cues that support disambiguation when visual evidence is ambiguous (Kim et al., 2024c). Transformer-based architectures discard external detectors and directly encode images into object and relation tokens through global attention, enhancing both detection and reasoning (Li et al., 2022b; Im et al., 2024). Despite these advances, predicates such as "standing on" and "riding" are still represented by static embeddings whose semantics remain insensitive to image-specific evidence.

We propose AlignG, a prototype feedback learning framework that bridges global predicate semantics with image-specific relational context. Unlike prior approaches, AlignG

---

[1]Department of Computer Engineering, Gachon University, Seongnam, Republic of Korea. Correspondence to: Chang Choi <changchoi@gachon.ac.kr>.

*Proceedings of the 43rd International Conference on Machine Learning*, Seoul, South Korea. PMLR 306, 2026. Copyright 2026 by the author(s).

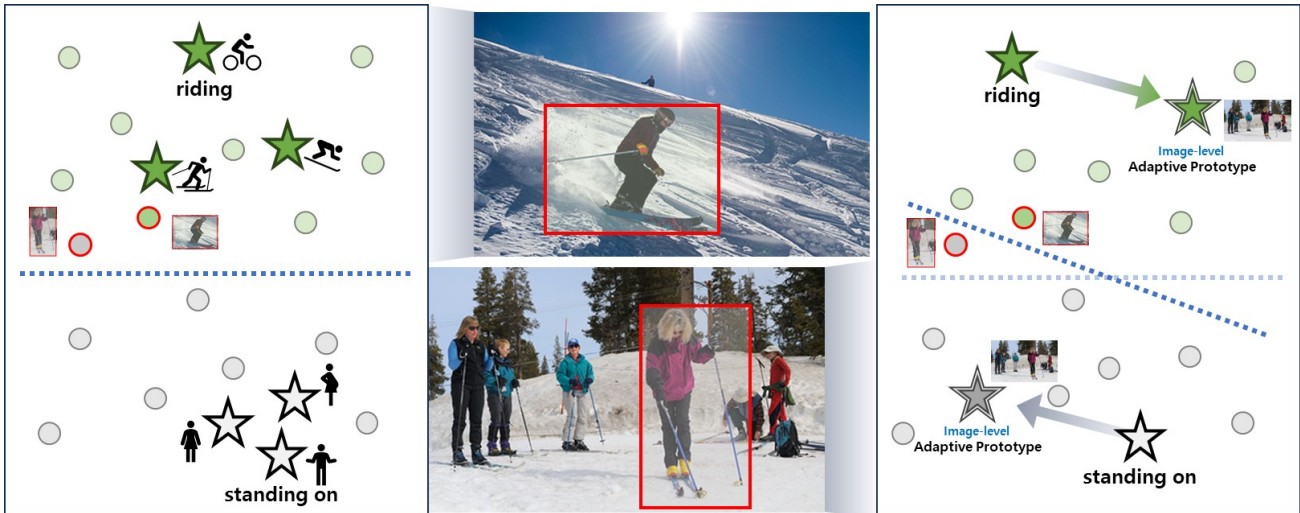

*Figure 1.* All individuals are preparing to ski, yet none are actively skiing. While they are all technically **"standing on"** skis, they are not **"riding"** them. This highlights the semantic gap between spatial and action-oriented predicates.

models mutual interactions between global prototypes and local relation candidates. This interaction enables a two-stage refinement where prototypes are conditioned on relational evidence to form context-aware semantics, and the adapted semantics are then fed back to recalibrate relation features. By integrating this prototype feedback into learning, AlignG allows predicate meanings to shift selectively with each image while maintaining the global semantic topology. As shown in Figure 1 (right), the framework helps distinguish predicates such as "standing on" and "riding", even when visual appearances are similar.

We evaluate AlignG on two widely used benchmarks, VG-150 and GQA-200, under PredCls, SGCls, and SGDet settings. The model achieves state-of-the-art performance, with F@100 gains of +1.4 on VG-150 and +2.7 on GQA-200 under SGDet. We further visualize per-image prototype similarity shifts and observe coherent context-dependent reorganization, where prototypes selectively merge or separate predicates according to scene evidence. Ablation studies further support that the gains stem from prototype feedback learning, which integrates image-specific evidence into predicate semantics. These findings highlight the importance of context-aware adaptation and suggest that the principle may extend to broader relational reasoning tasks.

Our main contributions are summarized as follows.

- We reformulate predicate learning in SGG as an image-conditioned adaptation problem, departing from the static-prototype paradigm of prior work.

- We introduce AlignG, a two-stage prototype feedback framework that adapts predicate prototypes to image-specific evidence and recalibrates relation features un-

der the adapted prototypes, refining ambiguous predicates while preserving global semantic coherence.

- We demonstrate state-of-the-art performance on VG-150 and GQA-200 benchmarks, with prototype similarity visualizations and ablation analyses supporting the effectiveness of context-conditioned grounding.

## 2. Related Work

### 2.1. Scene Graph Generation

SGG has evolved from early context modeling into more sophisticated relational reasoning frameworks. IMP (Xu et al., 2017) pioneered iterative message passing that jointly refines object and predicate predictions through recurrent context propagation. Neural Motifs (Zellers et al., 2018) introduced global context modules to capture higher-order dependencies among objects and relations. VC-Tree (Tang et al., 2019) proposed dynamic tree structures to capture hierarchical dependencies among objects. GPS-Net (Lin et al., 2020) advanced relational reasoning with permutation-invariant structures and direction-aware message passing. Transformer-based models broadened this line, with SGTR (Li et al., 2022b) employing global attention over object and relation tokens and EGTR (Im et al., 2024) shifting to edge-centric relational encoding. CooK (Kim et al., 2024a) leveraged commonsense co-occurrence knowledge to reduce long-tail bias, and ST-SGG (Kim et al., 2024b) enforced self-training with pseudo-labeling to mitigate annotation sparsity and long-term bias in static-image settings.

Complementary efforts address dataset bias and the long-tailed distribution of predicates through debiasing objec-

tives and structural regularization (Tang et al., 2020; Yu et al., 2021; Yoon et al., 2023). External knowledge and auxiliary signals further enrich predicate interpretation. GB-Net (Zareian et al., 2020) integrated commonsense knowledge graphs to regularize predicate distributions with structured priors. EOA (Chen et al., 2023) embedded ontology alignments directly into predicate learning to strengthen semantic control. HiKER-SGG (Zhang et al., 2024) organized predicates into hierarchical graphs to capture multiple abstraction levels. LLM4SGG (Kim et al., 2024c) leveraged large language models to improve triplet extraction and class alignment in weakly supervised setups. Despite these advances, most approaches treat predicate semantics as static embeddings or apply post hoc corrections, leaving predicate representations insensitive to image-conditioned relational evidence.

### 2.2. Prototype-Based and Multi-Prototype Methods

A complementary line of work organizes the predicate space through prototype learning. PE-Net (Zheng et al., 2023) introduced predicate-specific prototypes derived from semantic priors as fixed anchors for relation types. C-SGG (Chen et al., 2024) decomposed predicates into multiple sub-prototypes to capture usage diversity within a label. UP-Net (Jung & Choi, 2025) designed part-based anchors to improve fine-grained discrimination at localized regions. MCL (Lyu et al., 2025) expanded prototype capacity through multi-concept learning that models diverse semantic facets. RA-SGG (Yoon et al., 2025) employed retrieval-augmented memory to supplement rare or ambiguous relations during training.

The common limitation is that prototypes remain static after training. Even with multiple anchors, embeddings do not reorganize semantics to reflect the relational evidence of a new image. For example, "standing on" and "riding" may share visual features, yet their semantics diverge when motion or intent is implied. This mismatch highlights the gap between rigid anchors and the fluid, context-dependent nature of real scenes. Bridging this gap requires prototypes that adapt to image-specific evidence while preserving dataset-level semantic coherence. AlignG follows this principle by treating prototypes as context-conditioned variables that adapt to image-specific relational evidence, enabling selective semantic reorganization while preserving the global prototype structure.

## 3. Background

### 3.1. Scene Graph Generation

SGG represents an image as a graph of objects and their pairwise semantic relations (Krishna et al., 2017). Given an input image $I$, an object detector predicts $M$ object

nodes $\{v_i = (b_i, y_i)\}_{i=1}^M$, where $b_i \in \mathbb{R}^4$ is the bounding box and $y_i \in \{1, \ldots, C\}$ is the object category. In the relation prediction stage, subject–object pairs $(s_j, o_j)$ are combined to form $N$ triplets $\{(s_j, p_j, o_j)\}_{j=1}^N$, where each predicate label $p_j$ is classified into one of the predefined predicate categories. The resulting scene graph is defined as $\mathcal{G} = (V, E)$, with node set $V = \{v_i\}_{i=1}^M$ and edge set $E = \{(s_j, p_j, o_j)\}_{j=1}^N$. A central challenge lies in learning relation embeddings $\mathbf{e}_j \in \mathbb{R}^d$ that preserve dataset-level semantic coherence while adapting to the contextual semantics of individual images.

### 3.2. Static Predicate Prototypes in SGG

PE-Net (Zheng et al., 2023) introduced a prototype-based framework that structures the relation embedding space by associating each predicate with a semantic prototype. Each predicate category $r \in \{1, \ldots, R\}$ is associated with a word embedding $\mathbf{t}_r \in \mathbb{R}^{d'}$, which is projected through a learnable matrix $\mathbf{W}_p \in \mathbb{R}^{d \times d'}$ to obtain a static predicate prototype $\bar{\mathbf{p}}_r = \mathbf{W}_p \mathbf{t}_r \in \mathbb{R}^d$. Given a subject–object pair $(s, o)$, visual features $\mathbf{x}_s, \mathbf{x}_o \in \mathbb{R}^d$ and semantic embeddings $\mathbf{t}_s, \mathbf{t}_o \in \mathbb{R}^{d'}$ are fused into object representations $\mathbf{v}_s, \mathbf{v}_o \in \mathbb{R}^d$. A fusion function $F(\cdot, \cdot)$ then produces a relation embedding $\mathbf{e}_j = F(\mathbf{v}_s, \mathbf{v}_o) \in \mathbb{R}^d$, which is aligned with its corresponding predicate prototype $\bar{\mathbf{p}}_{p_j}$. Through this design, PE-Net enforces consistency between relation embeddings and predicate-level semantics, providing structural constraints on the embedding space and improving the robustness of predicate classification.

While effective in linking visual and semantic spaces, this approach keeps prototypes fixed once training is complete. Consequently, it cannot capture contextual variations within individual images, which is problematic for polysemous predicates whose meaning shifts with image-specific evidence, and for long-tail predicates that lack sufficient representation. To address these limitations, AlignG redefines prototypes as adaptive entities $\mathbf{p}_r^{(I)} \in \mathbb{R}^d$ that update dynamically in response to relational cues from the current image $I$, and uses the adapted semantics as prototype feedback to recalibrate relation embeddings, reconciling dataset-level semantic structure with local contextual variation.

## 4. Context-Conditioned Prototype Alignment

In this section, we present AlignG, a prototype feedback learning framework that models the interaction between global predicate prototypes and image-specific relational evidence. Conventional methods either freeze prototypes as static anchors or decompose them into multiple clusters, neither of which adapts to per-image evidence at inference. AlignG addresses this with a two-stage refinement. Prototypes are adaptively updated using relational cues, and

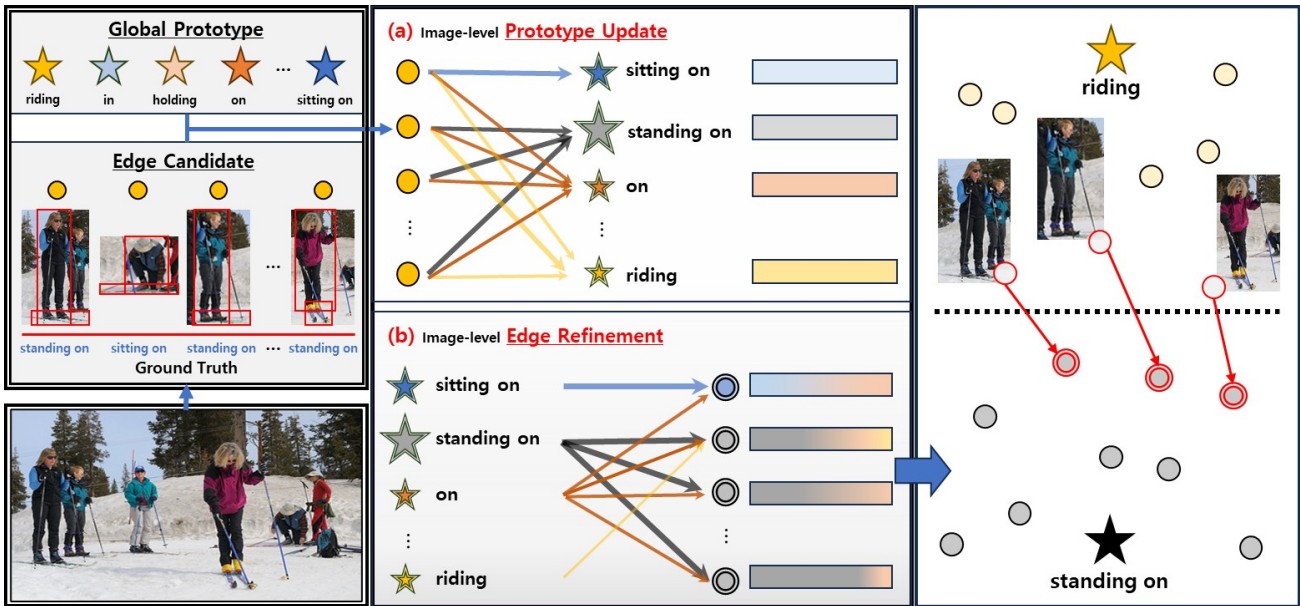

*Figure 2.* Overview of the AlignG framework. (a) Cross-Attention with a GRU produces image-conditioned prototypes from relation embeddings. (b) A reverse Cross-Attention propagates the adapted prototypes back to recalibrate each relation. The alignment loss is computed against the static prototypes to preserve global semantic structure.

relations are subsequently recalibrated under the guidance of these adapted prototypes. This design maintains the coherence of global semantics while tailoring them to the specifics of each image. An overview of the AlignG framework is illustrated in Figure 2.

### 4.1. Contextualizing Predicate Prototypes

Prototypes serve as dataset-wide anchors, but fixed anchors cannot accommodate polysemous or rare predicates. At the same time, directly overwriting them with relation-level signals would distort the global semantic structure. AlignG resolves this by updating prototypes incrementally, allowing them to integrate contextual information without losing stability.

To selectively integrate relational cues, AlignG aggregates relation embeddings for each prototype via compatibility-weighted cross-attention. This mechanism evaluates the compatibility between a prototype and all relation embeddings, assigning higher weights to the most relevant ones. Based on this formulation, the contextual signal for prototype $r$ is computed as

$$\mathbf{u}_r = \sum_{j=1}^{N} \frac{\exp\Big((\mathbf{W}_q\bar{\mathbf{p}}_r)^\top(\mathbf{W}_k\mathbf{e}_j)/\sqrt{d}\Big)}{\sum_{j'=1}^{N}\exp\Big((\mathbf{W}_q\bar{\mathbf{p}}_r)^\top(\mathbf{W}_k\mathbf{e}_{j'})/\sqrt{d}\Big)}\,\mathbf{W}_v\mathbf{e}_j,$$
(1)

where $\mathbf{W}_q, \mathbf{W}_k, \mathbf{W}_v \in \mathbb{R}^{d\times d}$ are learnable projection matrices. Through this aggregation, prototype $\bar{\mathbf{p}}_r$ selectively summarizes informative relation embeddings, producing a

contextual vector $\mathbf{u}_r \in \mathbb{R}^d$.

Once this contextual signal is obtained, it is incorporated into the prototype update process. This step ensures that prototypes evolve with local evidence while remaining stable anchors within the global semantic space. Formally, the update rule is given by

$$\mathbf{p}_r^{(I)} = \mathrm{GRUCell}(\mathbf{u}_r,\, \mathrm{LayerNorm}(\bar{\mathbf{p}}_r))\,,$$
(2)

where $\mathrm{LayerNorm}(\cdot)$ normalizes inputs for stable training. The resulting $\mathbf{p}_r^{(I)} \in \mathbb{R}^d$ serves as the image-specific refinement of prototype $r$, evolving under contextual evidence while preserving alignment with global semantics. The gated updater provides a controlled mechanism for adaptation, which mitigates abrupt semantic drift and promotes stable updates.

### 4.2. Recalibrating Relation Embeddings

Relation embeddings, in contrast to prototypes, are inherently transient. They exist only within the current image and do not persist across training. Their role is to represent context-specific semantics as precisely as possible, which motivates immediate recalibration rather than gradual refinement.

To address this, AlignG recalibrates relation embeddings under the guidance of adapted prototypes. A compatibility-weighted reverse cross-attention determines how much each adapted prototype contributes to refining a relation embedding. Formally, the prototype-informed feedback signal for

relation $j$ is computed as

$$\mathbf{u}_j = \sum_{r=1}^{R} \frac{\exp\left((\mathbf{W}'_q \mathbf{e}_j)^\top (\mathbf{W}'_k \mathbf{p}_r^{(I)})/\sqrt{d}\right)}{\sum_{r'=1}^{R} \exp\left((\mathbf{W}'_q \mathbf{e}_j)^\top (\mathbf{W}'_k \mathbf{p}_{r'}^{(I)})/\sqrt{d}\right)} \, \mathbf{W}'_v \mathbf{p}_r^{(I)},$$
(3)

where $\mathbf{W}'_q, \mathbf{W}'_k, \mathbf{W}'_v \in \mathbb{R}^{d \times d}$ are learnable projection matrices. This aggregation produces a guidance vector $\mathbf{u}_j \in \mathbb{R}^d$ for relation $j$.

After obtaining this feedback, each relation embedding is recalibrated by fusing its raw evidence with the prototype-informed correction. This single-step adjustment enables efficient adaptation without risk of error accumulation. The recalibration step is defined as

$$\tilde{\mathbf{e}}_j = f_{\text{proj}}([\text{LayerNorm}(\mathbf{e}_j); \mathbf{u}_j]),$$
(4)

where $f_{\text{proj}}(\cdot)$ is a projection network, $[\cdot; \cdot]$ denotes concatenation, and $\text{LayerNorm}(\cdot)$ stabilizes training. The resulting $\tilde{\mathbf{e}}_j \in \mathbb{R}^d$ captures both the raw relational evidence and the contextual guidance from prototypes, yielding a representation that is flexible to local variation yet consistent with global semantics.

### 4.3. Training Objectives and Prototype Adaptation

The optimization of AlignG follows the auxiliary objectives introduced in PE-Net (Zheng et al., 2023), which constrain the structure of the prototype space using regularization and alignment losses. Unlike previous methods, however, we do not introduce additional constraints, since our per-image prototype adaptation provides contextual flexibility without compromising semantic coherence.

To preserve the integrity of the prototype space, we adopt a regularization loss that prevents redundancy among prototypes while enforcing diversity. This encourages prototypes to remain informative anchors that span the semantic space. Formally, the prototype regularization loss is defined as

$$\mathcal{L}_{\text{reg}} = \underbrace{\frac{1}{R^2} \left\| \bar{\mathbf{P}}\bar{\mathbf{P}}^\top \right\|_{2,1}}_{\text{similarity penalty}}$$

$$+ \underbrace{\frac{1}{R} \sum_{r=1}^{R} \max\left\{ 0, \gamma_{\text{div}} - \min_{r' \neq r} \left\| \bar{\mathbf{p}}_r - \bar{\mathbf{p}}_{r'} \right\|_2^2 \right\}}_{\text{diversity constraint}}$$
(5)

where $\bar{\mathbf{P}} \in \mathbb{R}^{R \times d}$ stacks the $R$ prototypes by rows, and each row is $\ell_2$-normalized. The $\| \cdot \|_{2,1}$ term aggregates row-wise $\ell_2$ norms of the similarity matrix $\bar{\mathbf{P}}\bar{\mathbf{P}}^\top$, penalizing excessive inter-prototype similarity. The second term imposes a diversity margin $\gamma_{\text{div}}$ on pairwise squared prototype distances to encourage well-separated semantics.

Next, to ensure that relation embeddings remain aligned with their corresponding predicate prototypes, we apply a contrastive alignment loss. This loss pulls each relation embedding closer to its correct prototype while pushing it away from the most confusing negative. The alignment loss is defined as

$$\mathcal{L}_{\text{align}} = \max\left\{ 0, \| \tilde{\mathbf{e}}_j - \bar{\mathbf{p}}^+ \|_2^2 - \| \tilde{\mathbf{e}}_j - \bar{\mathbf{p}}^- \|_2^2 + \gamma \right\},$$
(6)

where $\bar{\mathbf{p}}^+$ is the prototype of the ground-truth predicate, $\bar{\mathbf{p}}^-$ is the closest non-target prototype to $\tilde{\mathbf{e}}_j$, and $\gamma$ is a margin hyperparameter. We intentionally compute the alignment loss against the static global prototypes $\bar{\mathbf{p}}_r$ rather than the adapted $\mathbf{p}_r^{(I)}$. This anchors the classifier to image-agnostic semantic centers, prevents trivial co-adaptation of relations and prototypes within the same image, and improves stability and generalization. Although $\mathcal{L}_{\text{align}}$ uses static prototypes $\bar{\mathbf{p}}_r$, both $\mathcal{L}_{\text{cls}}$ and $\mathcal{L}_{\text{align}}$ are computed on the recalibrated relations $\tilde{\mathbf{e}}_j$, so gradients propagate through relation recalibration and prototype adaptation.

Finally, the overall training objective integrates classification, regularization, and alignment losses

$$\mathcal{L} = \mathcal{L}_{\text{cls}} + \mathcal{L}_{\text{reg}} + \mathcal{L}_{\text{align}},$$
(7)

where $\mathcal{L}_{\text{cls}}$ is the standard cross-entropy loss supervising predicate classification. This combination ensures that AlignG maintains a globally coherent and structurally diverse prototype space, while dynamically adapting prototypes to relational evidence on a per-image basis.

## 5. Experiments

### 5.1. Datasets and Implementation Details

**VG-150.** The Visual Genome dataset contains 108,077 images annotated with objects and pairwise relations (Krishna et al., 2017). Following previous studies (Li et al., 2021; Jung et al., 2023; Lyu et al., 2025), we adopt the VG-150 split, which retains the 150 most frequent object categories and 50 predicate categories. Each image contains on average 38 objects and 22 relations, resulting in a dense annotation space with a long-tailed distribution. This benchmark has become a standard for evaluating scene graph generation models, as it poses challenges in both frequent and rare predicate recognition.

**GQA-200.** The GQA dataset was originally introduced for compositional visual reasoning (Hudson & Manning, 2019). More recently, it has also been adopted for SGG evaluation in previous studies (Jeon et al., 2024; Yoon et al., 2025), where the GQA-200 split, consisting of the 200 most common object categories and 100 predicate categories, has been widely used. Compared to VG-150, GQA-200 provides a broader set of relations and emphasizes fine-grained reasoning. The dataset contains images with more

*Table 1.* Performance comparison of AlignG and SOTA on VG-150. † indicates the use of dataset-level predicate frequency/co-occurrence statistics for long-tailed predicate handling.

| B | Method | PredCls | | | SGCls | | | SGDet | | |
|---|--------|---------|---|---|-------|---|---|-------|---|---|
| | | R@50/100 | mR@50/100 | F@50/100 | R@50/100 | mR@50/100 | F@50/100 | R@50/100 | mR@50/100 | F@50/100 |
| Specific | KERN† (Chen et al., 2019) CVPR'19 | 65.8 / 67.6 | 17.7 / 19.2 | 27.9 / 29.9 | 36.7 / 37.4 | 9.4 / 10.0 | 15.0 / 15.8 | 27.1 / 29.8 | 6.4 / 7.3 | 10.4 / 11.7 |
| | BGNN† (Li et al., 2021) CVPR'21 | 59.2 / 61.3 | 30.4 / 32.9 | 40.2 / 42.8 | 37.4 / 38.5 | 14.3 / 16.5 | 20.7 / 23.1 | 31.0 / 35.8 | 10.7 / 12.6 | 15.9 / 18.6 |
| | HetSGG† (Yoon et al., 2023) AAAI'23 | 57.8 / 59.1 | 31.6 / 33.5 | 40.9 / 42.8 | 37.6 / 38.7 | 17.2 / 18.7 | 23.6 / 25.2 | 30.0 / 34.6 | 12.2 / 14.4 | 17.3 / 20.3 |
| | SQUAT† (Jung et al., 2023) CVPR'23 | 55.7 / 57.9 | 30.9 / 33.4 | 39.7 / 42.4 | 33.1 / 34.4 | 17.5 / 18.8 | 22.9 / 24.3 | 24.5 / 28.9 | 14.1 / 16.5 | 17.9 / 21.0 |
| | CooK† (Kim et al., 2024a) ICML'24 | 62.1 / 64.2 | 33.7 / 35.8 | 43.7 / 46.0 | 39.1 / 40.0 | 17.5 / 18.6 | 24.2 / 25.4 | 30.1 / 34.6 | 12.6 / 14.9 | 17.8 / 20.8 |
| Motifs | Motifs (Zellers et al., 2018) CVPR'18 | 64.6 / 66.0 | 15.2 / 16.2 | 24.6 / 26.0 | 38.0 / 38.9 | 8.7 / 9.3 | 14.2 / 15.0 | 31.0 / 35.1 | 6.7 / 7.7 | 11.0 / 12.6 |
| | TDE (Tang et al., 2020) CVPR'20 | 46.2 / 51.4 | 25.5 / 29.1 | 32.9 / 37.2 | 27.7 / 29.9 | 13.1 / 14.9 | 17.8 / 19.9 | 16.9 / 20.3 | 8.2 / 9.8 | 11.0 / 13.2 |
| | NICE† (Li et al., 2022a) CVPR'22 | 55.1 / 57.2 | 29.9 / 32.3 | 38.8 / 41.3 | 33.1 / 34.0 | 16.6 / 17.9 | 22.1 / 23.5 | 27.8 / 31.8 | 12.2 / 14.4 | 17.0 / 19.8 |
| | IETrans† (Zhang et al., 2022) ECCV'22 | 54.7 / 56.7 | 30.9 / 33.6 | 39.5 / 42.2 | 32.5 / 33.4 | 16.8 / 17.9 | 22.2 / 23.3 | 26.4 / 30.6 | 12.4 / 14.9 | 16.9 / 20.0 |
| | CFA (Li et al., 2023) ICCV'23 | 54.1 / 56.6 | 35.7 / 38.2 | 43.0 / 45.6 | 34.9 / 36.1 | 17.0 / 18.4 | 22.9 / 24.4 | 27.4 / 31.8 | 13.2 / 15.5 | 17.8 / 20.8 |
| | ST-SGG† (Kim et al., 2024b) ICLR'24 | 53.9 / 57.7 | 28.1 / 31.5 | 36.9 / 40.8 | 33.4 / 34.9 | 16.9 / 18.0 | 22.4 / 23.8 | 26.7 / 30.7 | 11.6 / 14.2 | 16.2 / 19.4 |
| | DPL† (Jeon et al., 2024) ECCV'24 | 54.4 / 56.3 | 33.7 / 37.4 | 41.6 / 44.9 | 32.6 / 33.8 | 18.5 / 20.1 | 23.6 / 25.2 | 24.5 / 28.7 | 13.0 / 15.6 | 17.0 / 20.2 |
| | SSC-SGG† (Yang et al., 2025) AAAI'25 | 59.7 / 62.0 | 31.5 / 34.0 | 41.2 / 43.9 | 37.4 / 38.5 | 17.8 / 18.9 | 24.1 / 25.4 | 28.6 / 33.0 | 12.3 / 14.4 | 17.2 / 20.1 |
| PE-Net | PE-Net (Zheng et al., 2023) CVPR'23 | 64.9 / 67.2 | 31.5 / 33.8 | 42.4 / 45.0 | 39.4 / 40.7 | 17.8 / 18.9 | 24.5 / 25.8 | 30.7 / 35.2 | 12.4 / 14.5 | 17.7 / 20.4 |
| | MCL† (Lyu et al., 2025) IEEE TIP'25 | 59.3 / 61.8 | 40.2 / 42.4 | 47.9 / **50.3** | 36.2 / 37.5 | 23.3 / 24.8 | 28.4 / 29.9 | 27.3 / 31.7 | 14.9 / 17.3 | 19.3 / 22.4 |
| | RA-SGG† (Yoon et al., 2025) AAAI'25 | 62.2 / 64.1 | 36.2 / 39.1 | 45.7 / 48.6 | 38.2 / 39.1 | 20.9 / 22.5 | 27.0 / 28.6 | 26.0 / 30.3 | 14.4 / 17.1 | 18.5 / 21.9 |
| | **AlignG†** | 59.2 / 61.3 | **40.6 / 42.6** | **48.2** / 50.3 | 34.6 / 35.8 | **24.8 / 26.1** | **28.9 / 30.2** | 25.9 / 30.0 | **16.6 / 19.7** | **20.2 / 23.8** |

diverse contexts, allowing us to assess whether models can generalize beyond high-frequency relations and adapt to richer semantic structures.

**Implementation Details.** We follow the standard experimental setup used in previous studies (Zellers et al., 2018; Zheng et al., 2023; Yoon et al., 2025). A Faster R-CNN detector with a ResNeXt-101-FPN (Tang, 2020) backbone is employed for object proposals. The detector is pretrained and kept frozen during training, while relation prediction modules are optimized. Predicate semantics are initialized with 300-dimensional GloVe embeddings (Pennington et al., 2014). All experiments are implemented in PyTorch and conducted on RTX 4090 GPUs. Training is performed for 60k iterations with a batch size of 8, using SGD with learning rate $1 \times 10^{-3}$, momentum 0.9, and weight decay $1 \times 10^{-4}$. The diversity margin is set to $\gamma_{\text{div}} = 3.0$, and the alignment margin to $\gamma = 20.0$. We report results under two settings: standard training and a frequency-aware long-tailed protocol marked with †, following common practice in unbiased SGG.

## 5.2. Evaluation Settings

**SGG Tasks.** We follow the standard setup in previous studies (Li et al., 2021; Kim et al., 2024a; Yoon et al., 2025) to evaluate AlignG under three task configurations. Predicate Classification (PredCls) assumes that ground-truth bounding boxes and object categories are available, and the goal is to predict the predicate label for each subject–object pair. Scene Graph Classification (SGCls) also assumes ground-truth bounding boxes but requires both object categories and predicates to be inferred. Scene Graph Detection (SGDet) is the most challenging scenario, where the model must detect objects, assign categories, and simultaneously predict pairwise predicates directly from raw images.

**Evaluation Metrics.** We report Recall@K (R@K), Mean Recall@K (mR@K), and F@K. R@K measures the proportion of ground-truth triplets included in the top-K predictions, reflecting overall recall. mR@K averages per-class recall, alleviating the bias toward frequent predicates and highlighting performance under long-tailed distributions. F@K, the harmonic mean of Recall and Mean Recall, has been emphasized in recent studies (Yoon et al., 2025; Lyu et al., 2025) as a balanced indicator, since it reflects both global accuracy and fairness across predicate categories.

## 5.3. Comparisons with State-of-the-Art Methods

**Results on Visual Genome.** Table 1 summarizes the comparison of AlignG with previous state-of-the-art methods on VG-150. AlignG† achieves the highest mR@100 of 42.6 in PredCls, 26.1 in SGCls, and 19.7 in SGDet. Compared to the previous best model MCL†, this corresponds to gains of +0.2, +1.3, and +2.4, respectively. MCL† broadens predicate semantics through multi-concept learning, yet its expanded prototypes remain fixed at inference and cannot reorganize to reflect image-specific context, particularly under polysemous or rare predicates. RA-SGG† enhances diversity by retrieving external exemplars, yet the retrieved cues remain external to the current image rather than being derived from its own relational evidence. AlignG† instead enforces structural consistency between edge-level cues and adaptive prototypes, matching the best F@100 of 50.3 in PredCls, with leading scores of 30.2 in SGCls and 23.8 in SGDet, and the largest gain of +1.4 on SGDet despite a moderate R@100 reduction. Relative to the backbone PE-Net, AlignG† shows substantial gains of +8.8, +7.2, and +5.2 in mR@100 with minimal architectural change, confirming that the improvements stem from prototype feedback rather than concept expansion or external retrieval.

*Table 2.* Performance comparison of AlignG and SOTA on GQA-200. † indicates the use of dataset-level predicate frequency/co-occurrence statistics for long-tailed predicate handling.

| Method | PredCls | | | SGCls | | | SGDet | | |
|---|---|---|---|---|---|---|---|---|---|
| | R@50/100 | mR@50/100 | F@50/100 | R@50/100 | mR@50/100 | F@50/100 | R@50/100 | mR@50/100 | F@50/100 |
| VTransE (Zhang et al., 2017) CVPR'17 | 55.7 / 57.9 | 14.0 / 15.0 | 22.4 / 23.8 | 33.4 / 34.2 | 8.1 / 8.7 | 13.0 / 13.9 | 27.2 / 30.7 | 5.8 / 6.6 | 9.6 / 10.9 |
| VCTree (Tang et al., 2019) ICCV'19 | 63.8 / 65.7 | 16.6 / 17.4 | 26.4 / 27.5 | 34.1 / 34.8 | 7.9 / 8.3 | 12.8 / 13.4 | 28.3 / 31.9 | 6.5 / 7.4 | 10.6 / 12.0 |
| PE-Net (Zheng et al., 2023) CVPR'23 | 54.3 / 56.0 | 26.2 / 27.1 | 35.4 / 36.5 | 26.2 / 27.0 | 11.2 / 11.5 | 15.7 / 16.1 | 19.5 / 22.9 | 10.3 / 11.9 | 13.5 / 15.7 |
| DPL† (Jeon et al., 2024) ECCV'24 | 50.3 / 52.3 | 31.6 / 33.9 | 38.8 / 41.1 | 25.0 / 25.9 | 13.3 / 14.4 | 17.4 / 18.5 | 15.0 / 19.0 | 11.1 / 13.1 | 12.8 / 15.5 |
| RA-SGG† (Yoon et al., 2025) AAAI'25 | 48.3 / 50.1 | 35.4 / 36.8 | 40.9 / 42.4 | 19.9 / 20.8 | 16.4 / 17.2 | 18.0 / 18.8 | 16.3 / 19.0 | 12.9 / 15.0 | 14.4 / 16.8 |
| **AlignG**† | 48.9 / 50.7 | **36.6** / **37.9** | **41.8** / **43.4** | 23.7 / 24.6 | **16.7** / **17.4** | **19.6** / **20.4** | 22.7 / 26.3 | **13.6** / **15.5** | **17.0** / **19.5** |

*Table 3.* Ablation study of **AlignG** on VG-150. ✓ and $\mathcal{G}$ in Proto denote concatenation-based and gated recurrent prototype alignment. † indicates the use of dataset-level predicate frequency/co-occurrence statistics for long-tailed predicate handling.

| Method | Component | | PredCls | | | SGCls | | | SGDet | | |
|---|---|---|---|---|---|---|---|---|---|---|---|
| | Edge | Proto | R@50/100 | mR@50/100 | F@50/100 | R@50/100 | mR@50/100 | F@50/100 | R@50/100 | mR@50/100 | F@50/100 |
| **PE-Net** | | | 64.9 / 67.2 | 31.5 / 33.8 | 42.4 / 45.0 | 39.4 / 40.7 | 17.8 / 18.9 | 24.5 / 25.8 | 30.7 / 35.2 | 12.4 / 14.5 | 17.7 / 20.4 |
| **AlignG** | ✓ | | 63.1 / 65.3 | 33.8 / 36.1 | 44.0 / 46.5 | 40.9 / 41.8 | 16.9 / 18.2 | 23.9 / 25.4 | 30.4 / 34.7 | 12.9 / 15.1 | 18.1 / 21.0 |
| | ✓ | ✓ | 63.5 / 65.6 | 34.0 / 36.2 | 44.3 / 46.7 | 38.6 / 39.7 | 19.3 / 20.4 | 25.7 / 27.0 | 30.2 / 34.6 | 12.7 / 14.9 | 17.9 / 20.8 |
| | ✓ | $\mathcal{G}$ | 63.0 / 65.1 | **35.0** / **37.4** | **45.0** / **47.5** | 38.4 / 39.4 | **19.7** / **20.8** | **26.0** / **27.2** | 30.1 / 34.4 | **13.0** / **15.4** | **18.2** / **21.3** |
| **PE-Net**† | | | 59.0 / 61.4 | 38.8 / 40.7 | 46.8 / 48.9 | 36.1 / 37.3 | 22.2 / 23.5 | 27.5 / 28.8 | 26.5 / 30.9 | **16.7** / 18.8 | **20.5** / 23.4 |
| **AlignG**† | ✓ | $\mathcal{G}$ | 59.2 / 61.3 | **40.6** / **42.6** | **48.2** / **50.3** | 34.6 / 35.8 | **24.8** / **26.1** | **28.9** / **30.2** | 25.9 / 30.0 | 16.6 / **19.7** | 20.2 / **23.8** |

**Results on GQA-200.** Table 2 presents the results on GQA-200. AlignG† obtains 37.9 mR@100 in PredCls, 17.4 in SGCls, and 15.5 in SGDet, showing consistent improvements over RA-SGG† and +10.8 mR@100 over PE-Net. More importantly, in terms of F@100, AlignG† achieves 43.4 in PredCls, 20.4 in SGCls, and 19.5 in SGDet, yielding gains of +1.0, +1.6, and +2.7, respectively. Since GQA-200 emphasizes compositional and fine-grained reasoning, balancing R@K and mR@K becomes particularly challenging. The consistent improvements in F@100 highlight that AlignG† effectively reconciles this trade-off by aligning image-level relations with dataset-level prototypes, enabling robust recognition under diverse semantic contexts.

## 5.4. Ablation Study

**Component Analysis.** Table 3 summarizes ablations on VG-150. Starting from the PE-Net backbone, adding the edge update improves mR@100 by +2.3 in PredCls and +0.6 in SGDet, highlighting the benefit of modeling localized relational cues. Incorporating prototype alignment via concatenation further refines these representations, yielding +2.2 mR@100 in SGCls over the edge-only variant. Replacing the concatenation-based prototype update with a GRU consistently enhances mR@100 across tasks by +1.2 in PredCls, +0.4 in SGCls, and +0.5 in SGDet, and increases F@100 by up to +0.8, while maintaining comparable R@100. Appendix A provides a per-predicate breakdown of this ablation, and Appendix B compares the GRU against simpler update rules such as EMA and residual connections.

**Reweighting Sensitivity.** To examine the effect of frequency-based reweighting, we additionally apply it to

*Table 4.* Computational overhead of **AlignG** compared to the PE-Net backbone. F@100 is reported on VG-150 under SGDet.

| Method | FLOPs | Training | Inference | F@100 |
|---|---|---|---|---|
| **PE-Net** | 472.36G | 0.35 s/iter | 30.84 FPS | 20.4 |
| **AlignG** | 479.41G | 0.38 s/iter | 30.14 FPS | 21.3 |
| Δ | +7.05G | +0.03 s/iter | −0.70 FPS | +0.9 |

the PE-Net baseline. As shown in Table 3, reweighting alone improves mR@100 by +6.9 in PredCls, while our alignment mechanism achieves further gains even without relying on reweighting. When combined with reweighting, AlignG yields the best overall performance, demonstrating that our approach complements rather than depends on this strategy. These results indicate that aligning edge-level signals with global prototypes, particularly via a GRU-based updater, matches the R@100 of the reweighted PE-Net baseline while further boosting mR@100, offering an architectural alternative to recall-sacrificing reweighting.

**Computational Overhead.** The prototype-feedback mechanism in AlignG is designed to scale linearly with the number of relation candidates $P$ per image. Standard self-attention over $P$ candidates incurs $\mathcal{O}(P^2)$ cost, which becomes prohibitive in dense scenes. In contrast, AlignG performs cross-attention only between a fixed set of $R$ predicate prototypes, where $R = 50$ on VG-150, and the $P$ candidates, producing an $R \times P$ attention map. As reported in Table 4, AlignG adds only +7.05G FLOPs and +0.03 s per training iteration, while preserving real-time inference at over 30 FPS. Despite this minimal overhead, AlignG achieves a +0.9 F@100 gain over PE-Net under SGDet.

*Table 5.* Predicate-level confusion analysis on VG-150 under Pred-Cls. Numbers in the form "X (Y)" indicate *X* AlignG-resolved errors out of *Y* total PE-Net errors.

| GT | $\rightarrow$ | Confused | Resolved (Total) | Rate |
|---|---|---|---|---|
| lying on | $\rightarrow$ | laying on | 23 (54) | 42.6% |
| carrying | $\rightarrow$ | holding | 53 (197) | 26.9% |
| watching | $\rightarrow$ | looking at | 18 (84) | 21.4% |
| walking on | $\rightarrow$ | standing on | 7 (36) | 19.4% |
| riding | $\rightarrow$ | standing on | 6 (52) | 11.5% |

**Confusion Analysis.** A natural concern is that context-conditioned adaptation might blur boundaries between visually similar predicates. To examine this, we analyze PE-Net's hard-negative confusions on PredCls and measure how many AlignG resolves. As shown in Table 5, AlignG resolves 11.5% to 42.6% of PE-Net's errors, with the rate varying by the nature of the pair. Near-synonyms such as "lying on" and "laying on" are resolved at the highest rate of 42.6%, while hierarchical collapses into the more generic "standing on" show the lowest rates, reflecting the inherent difficulty of inferring motion or intent from static images. The reduction across all five categories confirms that context-conditioned adaptation sharpens rather than blurs ambiguous semantic boundaries.

**Qualitative Analysis.** Figure 3 visualizes per-image changes in predicate prototype similarity, highlighting how semantic neighborhoods reorganize under image-specific relational evidence. In (a), apparel-related relations such as ⟨shoe, on, woman⟩ and ⟨woman, wearing, shoe⟩ increase the similarity between "on" and "wearing", while spatial predicates such as "behind" remain moderately aligned with "on" due to consistent spatial overlap. In (b), the similarity between "has" and "on" increases in a possession-like context such as a clock tower, whereas "on" and "above" remain separated, and the model predicts ⟨tower, above, building⟩ rather than "on", reflecting a context-consistent vertical interpretation. In (c), activity-state predicates such as "sitting on" separate from "on", while contact-related semantics increase the similarity between "wearing" and "on", and the model predicts ⟨people, using, phone⟩ instead of ⟨people, on, phone⟩, capturing functional interaction. Overall, these similarity shifts provide qualitative evidence that prototype feedback induces coherent context-dependent semantic reorganization by selectively merging or separating predicate semantics according to the scene.

## 6. Discussion

AlignG reconciles dataset-level predicate structure and context variability through prototype feedback, shifting from static prototypes to context-conditioned semantics that adapt to image-specific relational evidence. Context-conditioned

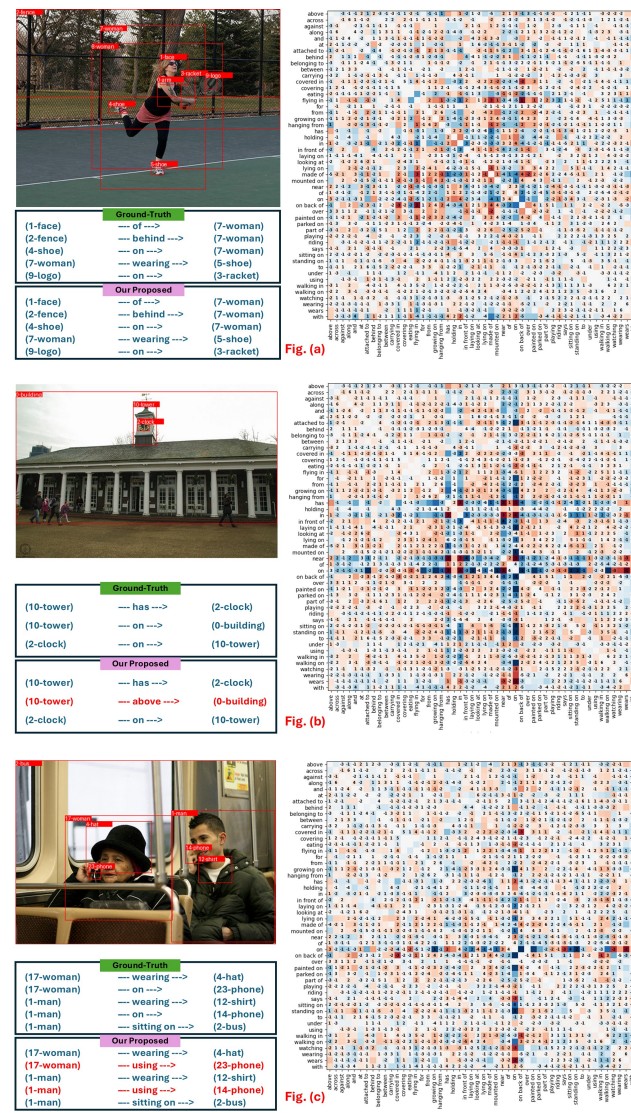

*Figure 3.* Context-conditioned predicate-similarity heatmaps for three scenes (a-c). Color encodes the change in prototype similarity. Red indicates an increase and blue indicates a decrease. Overlaid numerals denote the final per-image adaptive prototype similarity.

grounding benefits from aligning local relational signals with a shared semantic space, so that relational semantics remain coherent across the dataset yet sensitive to each image. However, per-image adaptation may be misled by contradictory contextual cues or disrupted by noisy detection, risking semantic drift toward local artifacts rather than meaningful relations. This exposes a tension between adaptability and stability, motivating safeguards such as confidence-aware evidence selection and stronger anchoring to global semantics to limit drift without suppressing context-dependent shifts. Improving this balance is an important direction, and the same principle of context-conditioned adaptation may extend to relational tasks such as fine-grained classification or dynamic graph learning.

# 7. Conclusion

We presented AlignG, a framework that learns context-conditioned predicate semantics via prototype feedback by aligning global predicate prototypes with relational cues from individual images. By integrating adaptability directly into predicate representations, AlignG demonstrates that semantic coherence and contextual flexibility can coexist within a unified structure. AlignG further shows that static prototypes and image-specific adaptation can operate together as complementary mechanisms, with static anchors providing semantic stability and adaptive prototypes capturing scene-specific evidence. While the current formulation is validated on image-based benchmarks, the same principle may extend to sequential settings such as video understanding, where relational semantics shift across both scenes and time. Incorporating temporal consistency into prototype feedback offers a promising path toward more temporally grounded scene interpretation.

## Acknowledgements

This work was supported in part by the National Research Foundation of Korea (NRF) grant funded by the Korea government (MSIT) (RS-2025-00559546, 45%), in part by the Basic Science Research Program through the National Research Foundation of Korea (NRF) funded by the Ministry of Education (RS-2024-00413584, 45%), and in part by the Institute of Information & Communications Technology Planning & Evaluation (IITP) grant funded by the Korea government (MSIT) under the Leading Generative AI Human Resources Development program (IITP-2026-RS-2026-25545512, 10%).

## Impact Statement

This paper advances scene graph generation, which supports downstream tasks such as visual question answering, image captioning, and robotic planning. AlignG inherits the annotation biases of Visual Genome and GQA, and per-image prototype adaptation could reinforce such biases when training correlations are spurious. We recommend validation on representative target distributions and human oversight when deploying AlignG in safety-sensitive applications such as surveillance, content moderation, or autonomous decision-making.

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

## A. Predicate-Level Analysis of Component Ablation

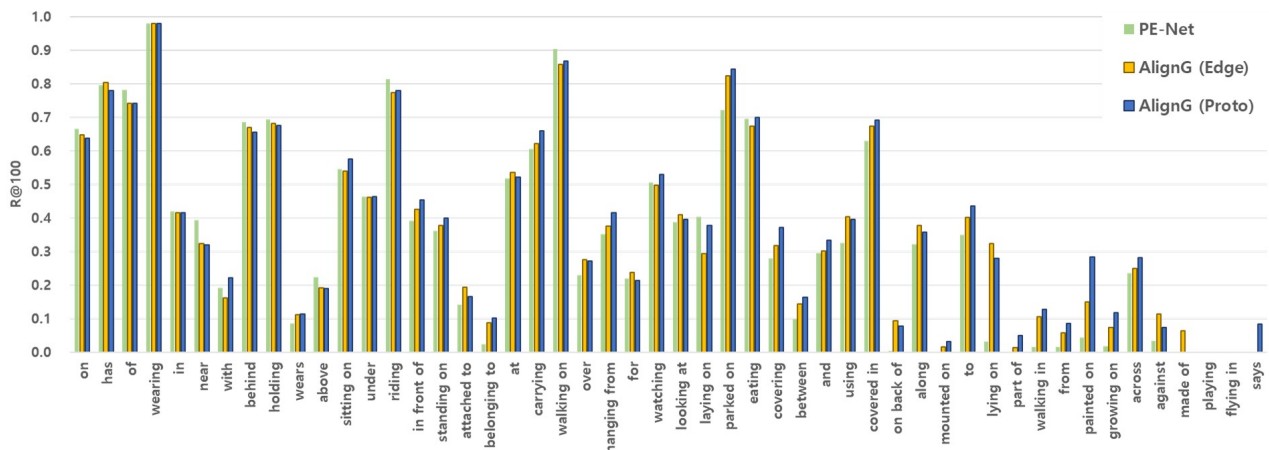

*Figure 4.* Per-predicate R@100 of PE-Net and AlignG variants on VG-150 PredCls. Predicates sorted by training frequency, head to tail.

To examine the per-predicate contributions of each component, Figure 4 reports R@100 on VG-150 PredCls for PE-Net and AlignG variants. The edge update primarily benefits predicates that can be inferred from strong local geometric or contact cues within a single subject–object pair, such as "lying on" and "on back of". In contrast, prototype feedback yields larger improvements on predicates whose interpretation depends on broader scene context or repeated relational patterns, including cases such as "painted on" and "says", where static prototypes tend to average over heterogeneous usages. Overall, the trends suggest complementary roles between localized relation evidence and context-conditioned prototype semantics, which is consistent with the design of AlignG.

## B. Comparison of Prototype Update Mechanisms

*Table 6.* Prototype update mechanisms on VG-150 PredCls. Variants differ only in the update rule of Eq. (2).

| Update Rule | R@100 | mR@100 | F@100 |
|---|---|---|---|
| **Identity (No Update)** | 69.06 | 30.70 | 42.52 |
| **Residual (Learnable)** | 68.99 | 31.12 | 42.90 |
| **Plain Addition** | 68.92 | 31.29 | 43.04 |
| **EMA (Learnable)** | 68.72 | 32.09 | 43.75 |
| **GRU (Ours)** | 65.06 | **37.36** | **47.47** |

To justify the necessity of the gated recurrent updater in AlignG, we compare it against four simpler update mechanisms on the PredCls task of VG-150. Each rule replaces only the GRU module in Eq. (2), keeping all other components fixed. Simpler rules such as Plain Addition or EMA linearly fuse incoming relational evidence into the prototype. In SGG, local visual evidence is inherently noisy, and linear merging causes severe semantic drift toward dominant head predicates: these variants achieve the highest R@100 of around 69 but suffer in mR@100, never exceeding 32.09. The GRU's gating mechanism actively filters irrelevant noise, enabling selective adaptation without overwriting the global semantic anchor. As a result, GRU boosts mR@100 by over $+5$ at the cost of a small $-3.7$ drop in R@100, achieving the highest F@100 of 47.47 and confirming that the gated update is structurally crucial for balancing global coherence with image-specific adaptation in long-tailed SGG.

## C. Sensitivity to Loss Weighting Coefficients

To examine the role of each loss component and the sensitivity of AlignG to their relative weights, we sweep the coefficients $\lambda_{sim}$, $\lambda_{div}$, and $\lambda_{align}$ on the PredCls task of VG-150. Here $\lambda_{sim}$ and $\lambda_{div}$ weight the similarity-penalty and diversity-constraint terms of $\mathcal{L}_{reg}$ in Eq. (5), respectively, and $\lambda_{align}$ scales $\mathcal{L}_{align}$ in Eq. (6); these are distinct from the diversity margin $\gamma_{div}$ and the alignment margin $\gamma$ inside Eqs. (5)–(6).

*Table 7.* Loss weighting ablation on VG-150 under PredCls. Baseline corresponds to unit weights.

| Group | # | $\lambda_{sim}$ | $\lambda_{div}$ | $\lambda_{align}$ | R@100 | mR@100 | F@100 |
|---|---|---|---|---|---|---|---|
| Baseline | 0 | 1.0 | 1.0 | 1.0 | 65.1 | **37.4** | **47.5** |
| A: Component | 1 | 0.0 | 1.0 | 1.0 | 69.6 | 29.6 | 41.5 |
|  | 2 | 1.0 | 0.0 | 1.0 | 70.1 | 26.6 | 38.6 |
|  | 3 | 1.0 | 1.0 | 0.0 | 62.0 | 32.6 | 42.7 |
| B: Sensitivity | 4 | 0.5 | 0.5 | 0.5 | 64.7 | 35.3 | 45.7 |
|  | 5 | 2.0 | 2.0 | 2.0 | 70.2 | 26.7 | 38.7 |
| C: Relative | 6 | 1.0 | 1.0 | 2.0 | 69.2 | 30.2 | 42.1 |
|  | 7 | 2.0 | 2.0 | 1.0 | 63.4 | 37.0 | 46.7 |

**Group A: Component.** Removing any of the three losses degrades performance through distinct failure modes. Without the similarity penalty or diversity constraint, R@100 inflates to roughly 70 while mR@100 drops below 30, indicating that the model defaults to dominant head predicates without a rigid anchor. Removing the alignment loss instead degrades R@100 to 62.0 as the classifier loses its tether to global semantic centers, lowering F@100 to 42.7.

**Group B: Sensitivity.** Over-penalizing all three losses anchors the model too strongly to global structure, suppressing image-specific adaptation and dropping F@100 to 38.7. Conversely, halving all weights loosens the global anchor and yields a small drop to 45.7.

**Group C: Relative.** Increasing $\lambda_{align}$ alone over-emphasizes the static centers and biases the model toward head classes, with F@100 falling to 42.1. Increasing the regularizers while keeping $\lambda_{align}$ fixed preserves balanced performance at 46.7. Across all configurations, the default $(1.0, 1.0, 1.0)$ Baseline achieves the best F@100 of 47.5, confirming that the loss terms in Eq. (7) are complementary and that AlignG remains stable under moderate perturbations.

# D. Robustness to Scene Density Variation

*Table 8.* Performance across scene density bins on VG-150 under PredCls and SGDet (mR@100). $G$ denotes the number of ground-truth relations per image, and $\Delta$ denotes AlignG's gain over PE-Net.

| Model | PredCls | | | |
|---|---|---|---|---|
|  | Very Sparse ($G \leq 3$) | Sparse ($3 < G \leq 10$) | Medium ($10 < G \leq 30$) | Dense ($G > 30$) |
| PE-Net | 32.85 | 33.86 | 33.40 | 38.94 |
| **AlignG** | **36.74** | **37.84** | **37.26** | **38.98** |
| $\Delta$ | +3.89 | +3.98 | +3.86 | +0.04 |

| Model | SGDet | | | |
|---|---|---|---|---|
|  | Very Sparse ($G \leq 3$) | Sparse ($3 < G \leq 10$) | Medium ($10 < G \leq 30$) | Dense ($G > 30$) |
| PE-Net | 15.46 | 14.14 | 14.12 | 18.86 |
| **AlignG** | **16.27** | **15.37** | **14.75** | **21.50** |
| $\Delta$ | +0.81 | +1.23 | +0.63 | +2.64 |

A natural concern is whether prototype adaptation degrades when the contextual signal in Eq. (1) becomes weak in sparse scenes. To examine this, we group the VG-150 test set by the number of ground-truth relations $G$ per image and evaluate mR@100 across four density bins for both PredCls and SGDet. Contrary to the intuition that adaptation might struggle when relations are sparse, AlignG yields strong gains precisely in Very Sparse scenes, with an 11.8% relative improvement on PredCls and +0.81 absolute gain on SGDet. This robustness arises from two complementary mechanisms. First, the Cross-Attention in Eq. (1) aggregates context from the entire dense proposal graph rather than from only the few ground-truth pairs, capturing the broader scene layout including background candidates. Second, when the aggregated context is uninformative, the gated updater in Eq. (2) acts as a pass-through that safely preserves the static prototype $\bar{p}_r$, selectively rejecting noisy updates rather than amplifying them. This noise suppression is particularly important in SGDet, where imperfect object detection produces unreliable relational cues, and explains why AlignG also achieves the largest gain of +2.64 in Dense scenes where redundant detections are most frequent.

# E. Qualitative Analysis of Context-Conditioned Adaptation

We present four qualitative scenes on the VG-150 PredCls task. Each case visualizes the subject-object pair along with surrounding triplets, showing how AlignG's Cross-Attention either corrects PE-Net's biases or over-adapts to misleading context. Figure 5 shows two success cases and Figure 6 shows two failure cases that motivate future work.

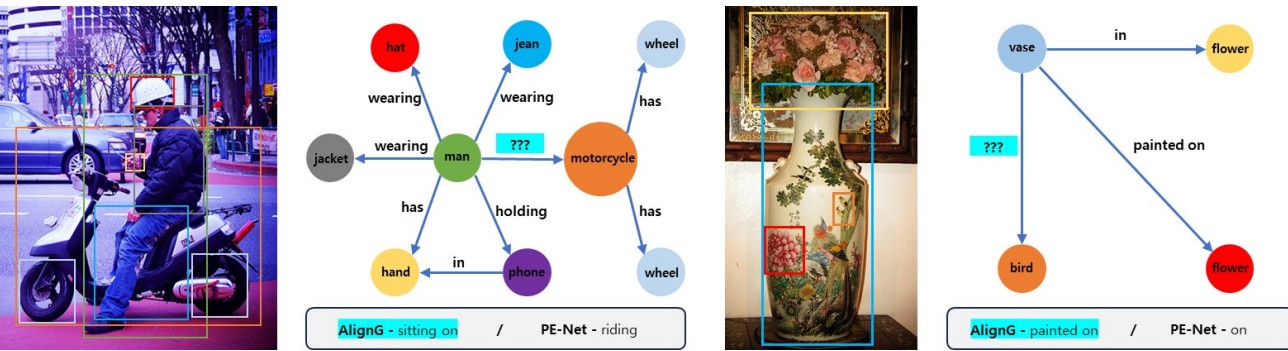

Figure 5. Qualitative success cases of AlignG on VG-150 PredCls: (left) man-motorcycle and (right) bird-vase.

**Figure 5 (left).** A man sits on a stationary motorcycle while using a phone. The ground-truth predicate is "sitting on". PE-Net predicts "riding", defaulting to the most frequent action label under the strong man-motorcycle co-occurrence bias. AlignG predicts the correct "sitting on" through the Cross-Attention in Eq. (1), aggregating the surrounding context including ⟨man, holding, phone⟩ and ⟨phone, in, hand⟩. The phone-in-hand cue shifts the predicate prototype toward a stationary posture and overrides the action-oriented language prior.

**Figure 5 (right).** A vase is decorated with bird and flower paintings on its surface, while real flowers are inserted from above. The ground-truth predicate is "painted on". PE-Net predicts the generic "on", falling back on a 3D contact interpretation without aggregating the surrounding context. AlignG predicts the correct "painted on" by aggregating the surrounding triplets ⟨flower, painted on, vase⟩ and ⟨flower, in, vase⟩. The decorative pattern shifts the prototype toward 2D surface graphics and recognizes the bird as part of the vase's painted decoration rather than a physical object resting on top.

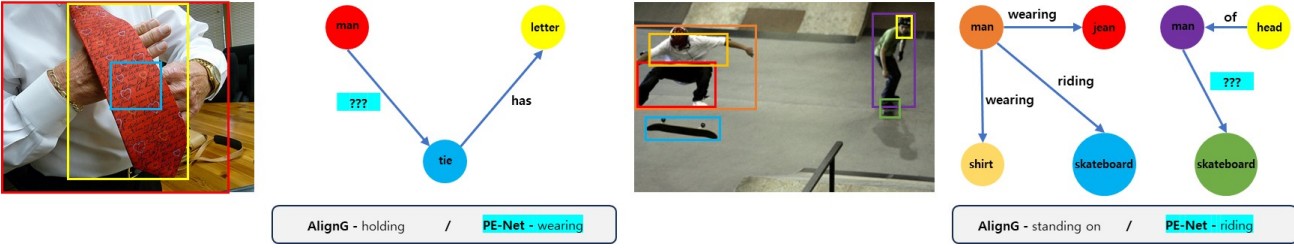

Figure 6. Qualitative failure cases of AlignG on VG-150 PredCls: (left) man-tie and (right) man-skateboard.

**Figure 6 (left).** A man lifts his tie from below with one hand, without grasping it. The ground-truth predicate is "wearing". PE-Net correctly predicts "wearing", whereas AlignG predicts "holding" from the visual proximity between hand and tie. Its Cross-Attention in Eq. (1) is misled by this proximity, while the only surrounding triplet ⟨tie, has, letter⟩ provides insufficient counter-evidence, downgrading the intrinsic apparel relation into a hallucinated interaction. Uncertainty-aware attention gating that suppresses low-confidence contextual signals is a promising direction to filter such misleading cues.

**Figure 6 (right).** A skater performs a mid-air trick on a skateboard, with a second skater visible in the background. The ground-truth predicate for the second skater's man-skateboard pair is "riding". PE-Net correctly predicts "riding" from the strong language prior, but AlignG over-adapts to local visual cues and predicts "standing on". Despite the supporting context ⟨man, riding, skateboard⟩ along with ⟨man, wearing, jean⟩, ⟨man, wearing, shirt⟩, and ⟨man, of, head⟩, its cross-attention over-weights the static visual posture of the skater and drifts toward a contact-based predicate that ignores the action context. A potential remedy is to integrate action-object affordance priors that resist semantic drift away from action-oriented predicates when motion cues are present.

