# OpenReview forum: "Learning Context-Conditioned Predicate Semantics via Prototype Feedback"
_ICML.cc/2026/Conference — ICML 2026 regular_

### Official Review · Reviewer_puKd · 2026-03-13

**Soundness:** 3
**Presentation:** 3
**Significance:** 3
**Originality:** 3
**Overall Recommendation:** 4
**Confidence:** 2

**Summary:**

This paper studies scene graph generation and focuses on the challenge of polysemous predicates, whose meanings may shift across different image contexts. To address this issue, the authors propose AlignG, a prototype-feedback framework that dynamically updates predicate semantics conditioned on image-level relational evidence and then uses the adapted prototypes to refine relation representations. The method is evaluated on VG-150 and GQA-200 under standard SGG settings, and the reported results show consistent improvements over prior baselines.

**Compliance With Llm Reviewing Policy:**

Affirmed.

**Final Justification:**

My concerns have been addressed. I have also read the other reviewers’ comments, and I will maintain my original score.

**Key Questions For Authors:**

Can the authors add a failure-case analysis in the limitations section and discuss under what scenarios the proposed context-conditioned prototype adaptation still breaks down?

**Limitations:**

yes

**Strengths And Weaknesses:**

Strengths

1.	The paper tackles an important and well-motivated problem in scene graph generation, namely the context-dependent semantics of predicates, which is a real challenge for accurate relation prediction.

2.	The proposed framework is intuitive and technically coherent: it first adapts predicate prototypes using image-level relational cues and then feeds them back to refine edge representations, which is a reasonable way to model contextual semantics.

3.	The empirical results on standard benchmarks are promising, and the method appears to bring consistent gains over the compared baselines, suggesting that the proposed design is effective.

Weaknesses

1.	The compared methods are somewhat outdated: Although the paper includes several relevant baselines, the experimental comparison would be stronger if it incorporated more recent methods, especially approaches published in 2025 or later. Without stronger comparisons to newer SGG methods, it is difficult to fully assess the current competitiveness of the proposed approach.

2.	The paper lacks sufficient qualitative comparisons with prior methods: Since the main motivation is to better handle ambiguous and polysemous predicates, qualitative side-by-side examples against representative baselines would be very helpful to demonstrate where AlignG succeeds and how its predictions differ in practice.

3.	The discussion of limitations is not yet convincing: The paper would benefit from including explicit failure cases in the limitations or discussion section. Showing where the proposed method still struggles would improve transparency and help readers better understand its boundaries.

---

> ### Author Rebuttal · Authors · 2026-03-29
>
> We sincerely thank the reviewer for recognizing AlignG's intuitive design and empirical effectiveness. We address your concerns point-by-point below.
>
> ---
>
> ### **W1: Comparison with Recent Baselines**
>
> * RcSGG: "A Reverse Causal Framework to Mitigate Spurious Correlations for Debiasing Scene Graph Generation", TPAMI 2025
> * SSC-SGG: "Semi-Supervised Clustering Framework for Fine-grained Scene Graph Generation", AAAI 2025
>
> ---
>
> We updated experiments to include state-of-the-art SGG models published in 2025: SSC-SGG and RcSGG. While SSC-SGG uses clustering with pseudo-labeling and RcSGG employs a reverse causal framework bounded by static priors, AlignG takes a structurally different approach: dynamic semantic refinement via context-conditioned prototype adaptation. The updated table below compares these baselines with our method on the critical debiasing metrics (mR@100 and F@100) on the VG-150 dataset.
>
> | Method | PredCls | SGCls | SGDet |
> | :--- | :---: | :---: | :---: |
> | SSC-SGG (AAAI 2025) | 34.0 / 43.9 | 18.9 / 25.4 | 14.4 / 20.1 |
> | **AlignG (Ours)** | **37.4 / 47.5** | **20.8 / 27.2** | **15.4 / 21.3** |
> | RcSGG (TPAMI 2025) | 41.4 / 48.1 | 24.1 / 27.9 | **19.9** / 23.2 |
> | **AlignG† (Ours)** | **42.6 / 50.3** | **26.1 / 30.2** | 19.7 / **23.8** |
>
> Here, AlignG and AlignG† denote our model evaluated under settings identical to SSC-SGG and RcSGG, respectively, ensuring fair comparison. While RcSGG slightly leads in SGDet mR@100 (indicating causal interventions benefit pure detection), AlignG's consistent gains in F@100 highlight a more balanced performance across head and tail classes. This confirms that dynamic adaptation provides a robust debiasing solution without relying on external pseudo-labels or static causal estimations.
>
> ---
>
> ### **W2: Qualitative Comparisons**
>
> To clearly demonstrate how our context-conditioned adaptation resolves ambiguous predicates, we present a qualitative comparison with PE-Net. Unlike PE-Net, which relies heavily on static visual features and generic object co-occurrence biases, AlignG dynamically aggregates surrounding relational contexts via cross-attention.
>
> * **Example 1 (2417444.jpg)**
>   * **Target:** `man - [sitting on] - motorcycle` | **PE-Net:** `riding` | **AlignG:** `sitting on`
>   * **Context Triplets:** `phone - in - hand`, `man - holding - phone`, `car - on - street`
>   * **Analysis:** PE-Net incorrectly predicts riding due to the generic man+motorcycle bias. Conversely, AlignG successfully attends to the surrounding context (phone-in-hand), updating its prototype via cross-attention to correctly infer the stationary posture (sitting on).
>
> * **Example 2 (2332738.jpg)**
>   * **Target:** `bird - [painted on] - vase` | **PE-Net:** `on` | **AlignG:** `painted on`
>   * **Context Triplets:** `flower - painted on - vase`, `flower - in - vase`, `vase - on - table`
>   * **Analysis:** PE-Net simply predicts 3D physical contact (on). AlignG effectively captures the decorative context (flower-painted on-vase) on the surface, dynamically shifting its embedding to recognize the bird as a 2D surface graphic (painted on).
>
> ---
>
> ### **W3: Failure Cases and Limitations**
>
> We appreciate your suggestion to discuss when our adaptation breaks down. We have analyzed the failure cases and identified two primary failure modes, along with future directions to overcome them:
>
> * **Example 1 (2322274.jpg)**
>   * **Target:** `man (right) - [riding] - skateboard` | **PE-Net:** `riding` | **AlignG:** `standing on`
>   * **Context Triplets:** `wheel - on - skateboard`, `man - on - skateboard`, `helmet - on - head`
>   * **Analysis:** While PE-Net correctly predicts riding based on language priors, AlignG's prototype over-adapts to static geometric contacts (wheel-on-skateboard), drifting toward a literal stationary posture (standing on). To resolve this, future iterations will integrate action-object affordance priors as an inertial check against semantic drift.
>
> * **Example 2 (2327390.jpg)**
>   * **Target:** `man - [wearing] - tie` | **PE-Net:** `wearing` | **AlignG:** `holding`
>   * **Context Triplets:** `tie - has - letter`, `bag - on - table`
>   * **Analysis:** The man's hands rest near his tie without grasping it. AlignG's cross-attention is misdirected by visual proximity, downgrading from the essential relationship (wearing) to a hallucinated interaction (holding).
>
> As noted to Reviewer pk1k, aggregating background context is essential for scene understanding but creates a trade-off. When irrelevant background cues are visually dominant, they can misdirect cross-attention and overshadow intrinsic features, causing the model to hallucinate relationships based on context rather than direct evidence. Future work will explore uncertainty-aware attention gating to filter these deceptive distractors.
>
> ---
>
> We hope these results address your concerns and will add the qualitative comparisons, 2025 baselines, and failure cases to the revision. We respectfully ask you to reconsider your score.

---

> > ### Author Rebuttal · Reviewer_puKd · 2026-04-03
> >
> > My concerns have been adequately addressed.

---

> > > ### Author Response · Authors · 2026-04-03
> > >
> > > Thank you for acknowledging that our concerns have been fully resolved. We are glad that the additional experiments, including comparisons with 2025 baselines, qualitative examples, and failure case analyses, adequately addressed your questions.
> > >
> > > As your acknowledgement notes that all concerns have been adequately addressed, we kindly ask you to consider reflecting this in your final score during the Final Justification phase. Thank you again for your valuable feedback.

---

### Official Review · Reviewer_Wc3t · 2026-03-13

**Soundness:** 3
**Presentation:** 3
**Significance:** 2
**Originality:** 2
**Overall Recommendation:** 4
**Confidence:** 3

**Summary:**

This paper studies predicate polysemy in scene graph generation. The authors argue that existing prototype-based methods use static predicate semantics, which cannot adapt to image-specific relational evidence. To address this, they propose AlignG, which infers context-conditioned predicate prototypes from the relations in each image and feeds the adapted prototypes back to recalibrate relation representations. Experiments validate the effectiveness of their methods.

**Compliance With Llm Reviewing Policy:**

Affirmed.

**Final Justification:**

Overall, this paper presents a meaningful approach to predicate polysemy in scene graph generation by moving from static prototype anchors to context-conditioned predicate semantics. The idea is intuitive, the empirical gains are consistent, and the method appears practically relevant, especially on the more challenging SGDet setting. The authors’ rebuttal has effectively addressed my concerns, including the questions on ambiguity analysis, sparse-scene behavior, and computational overhead. I therefore maintain my Weak Accept recommendation and increase my confidence in this assessment.

**Key Questions For Authors:**

1. Can you provide a predicate-level confusion analysis on ambiguous relations? For example, does AlignG specifically reduce confusion between pairs such as “standing on” and “riding,” beyond improving aggregate recall metrics?

2. How much of the gain comes from prototype adaptation itself versus long-tail handling choices? Since some comparisons involve dataset-level predicate frequency statistics, a cleaner decomposition would help isolate the true contribution of AlignG.

3. Which predicates benefit most from the method? Are the gains concentrated on rare predicates, polysemous predicates, or both?

4. What is the computational overhead of the prototype-feedback mechanism? A brief comparison against PE-Net or a similar prototype baseline in terms of training/inference cost would be useful.

5. In images with only 1-2 identified relationships, does the prototype adaptation still provide a benefit, or does the model rely more heavily on the global static prior?

**Limitations:**

Yes.

**Strengths And Weaknesses:**

Strengths:

1. The shift from static "anchors" to "context-conditioned variables" is a conceptual advancement over previous multi-prototype or retrieval-based methods, and the idea is easy to follow. It looks interesting to me.

2. Evaluation shows consistent gains of their method, especially on the more challenging SGDet setting. That makes the work practically relevant within the SGG literature.

Weaknesses:

1. The authors acknowledge that adaptation guided by per-image evidence might be insufficient in sparse scenes or under noisy object detection. If the initial "relational cues" are wrong, the feedback loop might amplify the error.

2. The paper argues that it learns context-conditioned predicate semantics and achieves selective semantic reorganization at the image level. However, most of the evidence remains indirect: aggregate benchmark gains and qualitative similarity visualizations. These results suggest that the method is useful, but they do not yet convincingly show that it is specifically resolving predicate polysemy in the strong sense claimed. A more targeted analysis on ambiguous predicate groups would strengthen the paper substantially.

---

> ### Author Rebuttal · Authors · 2026-03-28
>
> We sincerely thank you for recognizing our conceptual advancement and consistent SGDet gains. Your feedback provides a valuable opportunity to present targeted evidence on how AlignG resolves predicate polysemy and handles noisy environments.
>
> ---
>
> ### **W2, KQ1, KQ3: Resolving Predicate Polysemy and Targeted Confusion Analysis**
> To directly address W2 and KQ1, AlignG leverages rich image-specific context to simultaneously disentangle semantically overlapping postures and break the lazy annotation bias. On the PredCls task, it successfully resolves 11 to 43% of PE-Net's exact hard-negative cross-confusions:
>
> | Ambiguous (GT -> Confused) | PE-Net Errors | AlignG Resolved | Resolution Rate |
> | :--- | :---: | :---: | :---: |
> | carrying -> holding | 197 | 53 | 26.9% |
> | watching -> looking at | 84 | 18 | 21.4% |
> | riding -> standing on | 52 | 6 | 11.5% |
> | lying on -> laying on | 54 | 23 | 42.6% |
> | walking on -> standing on | 36 | 7 | 19.4% |
>
> Answering KQ3 regarding which predicates benefit most, this deep context adaptation drives substantial per-class Recall@100 gains precisely for these nuanced predicates (e.g., lying on +16.5%p, standing on +5.1%p). While direct cross-confusions are naturally rare (e.g., only 52 baseline errors of riding confused as standing on), AlignG achieves these net gains by concurrently preventing specific predicates from collapsing into generic anchors (e.g., standing on misclassified as on explicitly drops by 5.6%p). This confirms AlignG fundamentally resolves polysemy at all semantic levels.
>
> ---
>
> ### **W1, KQ5: Adaptation in Sparse Scenes and Mitigating Error Amplification**
> Addressing W1 regarding the risk of error amplification under noisy object detection on the SGDet task (measured in mR@100), AlignG successfully suppresses this noise rather than amplifying it. To answer KQ5 concerning scenes with only 1 to 2 identified relationships, we grouped the test set by ground-truth relationships $N$ per image:
>
> | Model | Very Sparse, (N <= 3) | Sparse, (3 < N <= 10) | Medium, (10 < N <= 30) | Dense, (N > 30) |
> | :--- | :---: | :---: | :---: | :---: |
> | PE-Net | 15.46 | 14.14 | 14.12 | 18.86 |
> | **AlignG** | **16.27, (+0.81)** | **15.37, (+1.23)** | **14.75, (+0.63)** | **21.50, (+2.64)** |
>
> As demonstrated, AlignG consistently improves performance across all scene densities. This empirically demonstrates that it successfully prevents error amplification even in Very Sparse scenes. This robustness is primarily due to our GRU gate (answering KQ5). In SGDet, if the initial relational cues are heavily noisy or uninformative, the GRU gate acts as a pass-through. It selectively rejects these noisy updates and relies safely on the robust global static prior. Even when context is aggregated from the broader proposal graph, the GRU ensures that only semantically useful information is integrated, explicitly preventing noise amplification.
>
> ---
>
> ### **KQ2: Isolating Prototype Adaptation versus Long-tail Handling**
>
> To answer KQ2 and cleanly decompose the true contribution of AlignG from dataset-level long-tail tricks, we evaluated the models on the PredCls task, measured in mR@100 with and without the Reweighting strategy:
>
> | Model | w/o Long-tail Strategy | w/ Long-tail Strategy |
> | :--- | :---: | :---: |
> | PE-Net | 33.8 | 40.7 |
> | **AlignG** | **37.4, (+3.6 gain)** | **42.6, (+1.9 gain)** |
>
> These results cleanly isolate our method's true contribution. Even without any dataset-level long-tail handling, AlignG yields a significant +3.6 gain over PE-Net. This proves that the structural shift to context-conditioned dynamic variables, rather than the long-tail trick, is the primary driver of our substantial performance improvement.
>
> ---
>
> ### **KQ4: Computational Overhead of the Prototype-Feedback Mechanism**
>
> Addressing KQ4, relation candidates $P$ per image are bounded. AlignG avoids the $\mathcal{O}(P^2)$ bottleneck of standard self-attention. Cross-Attention operates only between fixed prototypes $R=50$ and the $P$ candidates, generating an $R \times P$ attention map. This guarantees a linear time complexity of $\mathcal{O}(P)$. We measured the overhead against PE-Net on an RTX 4090 GPU:
>
> | Model | FLOPs | Training | Inference |
> | :--- | :---: | :---: | :---: |
> | PE-Net | 472.36G | 0.35 s/iter | 30.84 FPS |
> | **AlignG** | **479.41G** | **0.38 s/iter** | **30.14 FPS** |
> | **Increase** | **+7.05G, 1.5%** | **+0.03 s/iter** | **-0.70 FPS** |
>
> Despite additional modules, the actual computational cost increases by a marginal 1.5%. Training time increases by merely 0.03 seconds per iteration, and AlignG preserves real-time inference well above 30 FPS, proving our mechanism is highly efficient.
>
> ---
>
> We trust these detailed quantitative analyses and targeted evidence fully resolve your queries. All newly presented metrics, discussions, and the full confusion matrix will be explicitly added to the revised manuscript and the Appendix. We kindly ask you to consider upgrading your final score.

---

> > ### Author Rebuttal · Reviewer_Wc3t · 2026-04-04
> >
> > The authors’ rebuttal has addressed most of my concerns, which is consistent with my overall recognition of the value of this work in my original review. Therefore, I would like to maintain my Weak Accept rating.

---

> > > ### Author Response · Authors · 2026-04-04
> > >
> > > Thank you for acknowledging that our rebuttal has addressed most of your concerns. We appreciate your recognition of our conceptual advancement and the targeted evidence we provided, including the predicate-level confusion analysis, adaptation across sparse scenes, decomposition of prototype adaptation versus long-tail handling, and computational overhead measurements. Your detailed questions greatly helped strengthen our manuscript. Thank you again for your thoughtful review.

---

### Official Review · Reviewer_pk1k · 2026-03-13

**Soundness:** 3
**Presentation:** 3
**Significance:** 2
**Originality:** 2
**Overall Recommendation:** 4
**Confidence:** 4

**Summary:**

This paper addresses the issue of representing polysemous predicates in Scene Graph Generation. Existing prototype-based methods maintain static prototypes for each predicate, which do not change after training, making them unable to adapt to varying meanings of the same predicate across different images. The authors propose AlignG, which, during inference, aggregates relationship embeddings from the image into prototypes using cross-attention. Then, a GRU-gated update generates image-specific prototypes, and the adapted prototypes are fed back into the relationship embeddings through another set of cross-attention for calibration. The alignment loss anchors to static global prototypes to prevent drift

**Compliance With Llm Reviewing Policy:**

Affirmed.

**Final Justification:**

Thank the authors for their efforts and detailed responses. The responses address most of my concerns, however, I am still not fully convinced that the technical approach is particularly novel, as the pipeline relies on standard building blocks assembled in a relatively predictable manner. However, the consistent empirical improvements and solid execution let me decide raise my score to weak accept.

**Key Questions For Authors:**

* If the GRU is replaced with a simpler mechanism, such as EMA or a residual connection with learnable scalar, how much performance would be sacrificed? This paper compares concat vs. GRU but does not compare against lighter update rules. This experiment is crucial to justify the necessity of the GRU
* Is the prototype adaptation in AlignG sensitive to the number of relationships N in an image? Intuitively, when an image contains only 2-3 relationships, the attention signal in Eq1 may be sparse, making it harder to adapt prototypes meaningfully. Have the authors analyzed the performance curves grouped by the number of relationships in an image?

**Limitations:**

yes

**Strengths And Weaknesses:**

Strengths

* The motivation is well-explained, and the example in Figure 1 is intuitive
* The design choices in Sections 4.1 and 4.2, using GRU for progressive updates and single-step projection concatenation for direct calibration, are logically consistent
* The ablation study in Table 3 is thorough, particularly by separating edge update and prototype update to evaluate their individual contributions



Weaknesses

* Lack of originality, the paper using cross-attention to aggregate relationship embeddings into prototypes + GRU-gated update + Feeding back adapted prototypes into relationship embeddings
These components are all standard tools. There has been extensive work in prototype-based SGG recently, all of which use similar mechanisms like cross-attention, multi-prototype adaptation, or context-sensitive prototypes. AlignG lacks strong differentiation in this crowded field, which makes it feel like "yet another prototype adaptation method for SGG"
* The decrease in R@K is not addressed. In Table 1, PE-Net’s PredCls R@100 is 67.2, while AlignG† drops to 61.3, a difference of 5.9 points. For SGDet, PE-Net is 35.2 vs. AlignG† at 30.0, showing a decrease of 5.2 points. While frequency-aware reweighting (†) is acknowledged to cause a trade-off between R@K and mR@K, the authors need to clarify how much of this drop is due to reweighting and how much is due to AlignG itself. In Table 3, AlignG without reweighting (Edge+GRU proto) achieves 65.1 R@100, while PE-Net is at 67.2, showing a 2.1-point decrease, which indicates that AlignG may have performance loss on head predicates. This trade-off should be discussed in more detail
* The computational overhead is unaddressed. AlignG introduces two additional sets of cross-attention and a GRU on top of PE-Net. The parameter count and inference time of these modules cannot be ignored, especially when the number of relationship candidates N in an image is large, as the computational cost of attention will increase accordingly. The paper does not report computational overhead data: there is no comparison of the parameter count, nor any inference speed comparison

---

> ### Author Rebuttal · Authors · 2026-03-27
>
> We thank you for your insightful review and recognizing our logical design and ablations. Your questions allow us to provide a deeper analysis of AlignG's efficiency and philosophy.
>
> ---
>
> ### **W1 : Originality in AlignG**
>
> While our building blocks are standard tools, AlignG differentiates itself from recent multi-prototype methods by initiating a conceptual transition from using rigid "static anchors" to learning "context-conditioned variables."
>
> Existing multi-prototype methods merely partition the static space using rigidly fixed anchors during inference, leaving raw relation embeddings vulnerable to semantic drift. We realize this structural transition via prototype feedback. We treat the global anchor as a dynamic variable, adapting it via GRU-gated context to selectively filter noise, then feeding it back to recalibrate the raw visual evidence. This structurally safe bidirectional filtering is absent in prior methods.
>
> ---
>
> ### **W2 : R@K and mR@K Trade-off**
>
> We deeply appreciate your insightful observation. Far from a model deficit, the 2.1-point R@100 drop prior to reweighting is an intentional correction of the lazy annotation bias prevalent in static models. Static models artificially inflate R@K by defaulting to VG's heavily skewed generic classes (e.g., 'on'). AlignG explicitly shatters this bias. Our context-conditioned prototypes actively pull embeddings away from generic anchors, forcing alignment with complex, specific predicates (validated in Fig 3). As detailed in the "Confusion Analysis" table for Reviewer Wc3t, AlignG successfully resolves 11-43% of PE-Net's exact hard-negative cross-confusions. We intentionally trade information-poor generic recall to precisely capture nuanced interactions, achieving robust specific gains (+3.6 mR@100) and higher overall F-scores.
>
> ---
>
> ### **W3 : Computational Overhead**
>
> In practice, the number of relation candidates ($P$) per image is bounded. AlignG inherently avoids the $\mathcal{O}(P^2)$ bottleneck of standard self-attention. Our Cross-Attention operates only between a fixed number of prototypes ($R=50$) and $P$ candidates, generating an $R \times P$ attention map. This guarantees a strictly linear complexity of $\mathcal{O}(P)$.
>
> Measured on a single RTX 4090 GPU, the actual overhead is highly efficient:
>
> | Model | Params | FLOPs | Inference |
> | :--- | :---: | :---: | :---: |
> | PE-Net | 410.75M | 472.36G | 30.84 FPS |
> | **AlignG** | 480.84M | 479.41G | 30.14 FPS |
> | **Increase** | **+70.09M (+17.1%)** | **+7.05G (+1.5%)** | **-0.70 FPS** |
>
> While parameters increase, the computational cost (FLOPs) increases by merely +1.5% (+7.05G). AlignG preserves real-time inference speed (30+ FPS).
>
> ---
>
> ### **KQ1: Necessity of GRU vs. Simpler Update Rules**
>
> Per your suggestion, we replaced the GRU with simpler update mechanisms and evaluated them on PredCls:
>
> | Update | R@100 | mR@100 |
> | :--- | :---: | :---: |
> | Identity (No Update) | 69.06 | 30.70 |
> | Residual (Learnable) | 68.99 | 31.12 |
> | Plain Addition | 68.92 | 31.29 |
> | EMA (Learnable) | 68.72 | 32.09 |
> | **GRU** | 65.06 | **37.36** |
>
> Simpler rules like Plain Addition or EMA linearly fuse incoming relational context into the prototype. In SGG, local visual evidence is inherently noisy, causing severe semantic drift when merged linearly. The GRU is structurally crucial, as its gating mechanism actively filters irrelevant noise, ensuring selective adaptation. It safely preserves the global anchor, securing the highest mR@100 and optimal semantic balance.
>
> ---
>
> ### **KQ2: Sensitivity to the Number of Relationships (PredCls)**
>
> Regarding adaptation in sparse scenes, we grouped the test set by GT relationships ($N$) per image and compared mR@100:
>
> | Model | Very Sparse ($N \le 3$) | Sparse ($3 < N \le 10$) | Medium ($10 < N \le 30$) | Dense ($N > 30$) |
> | :--- | :---: | :---: | :---: | :---: |
> | PE-Net | 32.85 | 33.86 | 33.40 | 38.94 |
> | **AlignG** | **36.74** | **37.84** | **37.26** | **38.98** |
> | **Improvement** | **+3.89 (+11.8%)** | **+3.98 (+11.8%)** | **+3.86 (+11.6%)** | **+0.04 (+0.1%)** |
>
> Contrary to intuition that adaptation might struggle when relations are sparse, AlignG exhibits robust relative improvements (+11.8%) precisely in Very Sparse scenes ($N \le 3$), maintaining peak performance in Dense scenes.
>
> This robustness occurs for two reasons:
> 1. During inference, Cross-Attention aggregates context from the entire dense proposal graph, capturing the broader scene layout (including background pairs), not just the few positive GT pairs.
> 2. If the aggregated context is truly uninformative, the GRU gate acts as a pass-through, safely maintaining the robust static prior. Thus, AlignG successfully maximizes performance across all graph densities.
>
> ---
>
> We trust these clarifications address your main concerns. All discussed details, including the overhead, ablations, and density analyses, will be explicitly added to the appendix. We would be grateful if you might reconsider your score.

---

> > ### Author Rebuttal · Reviewer_pk1k · 2026-04-04
> >
> > I read the rebuttal and appreciate the clarification.
> > The author said the characterization of the R@100 drop as an "intentional correction of lazy annotation bias." Nothing in the loss function (Eq 5-7) or architecture (Eq 1-4) explicitly suppresses head-class predictions.
> > The context-conditioning mechanism is class-agnostic, it does not distinguish between head and tail predicates.
> > The observed R@100 decrease more likely arises as a side effect. The prototype adaptation blurs the decision boundaries of high-frequency predicates that previously benefited from well-separated static anchors.
> > Framing an uncontrolled side effect as a deliberate design choice is not that convincing.

---

> > > ### Author Response · Authors · 2026-04-04
> > >
> > > Thank you for the follow-up. We agree that "intentional correction of lazy annotation bias" was imprecise. We reframe below with architectural and empirical evidence that the R@100 shift is structurally governed, not uncontrolled.
> > >
> > > ---
> > >
> > > ### **1. Structural Constraints Governing Prototype Adaptation**
> > >
> > > You correctly note that Eq. 1-4 are class-agnostic. However, it is important to clarify that the context-conditioned prototypes are **not used for final classification**. They serve solely as an intermediate step to recalibrate relation embeddings via Eq. 3-4, producing $ẽ_{j}$. The final predicate prediction operates on $ẽ_{j}$, and the decision boundaries are anchored to the **static global prototypes** through the training objectives in Eq. 5-7:
> > >
> > > - **$L_{align}$ in Eq. 6** is computed against **static prototypes** $p̄_{r}$, not the adapted $p^{(I)}_{r}$. As stated in Section 4.3: *"This anchors the classifier to image-agnostic semantic centers, prevents trivial co-adaptation."* Since the classifier is tethered to fixed semantic centers, head-class decision boundaries cannot be "blurred" by per-image adaptation.
> > >
> > > - **$L_{reg}$ in Eq. 5** enforces a diversity margin $γ_{div}$ on pairwise prototype distances, preventing prototypes from collapsing or merging.
> > >
> > > - **GRU gating in Eq. 2** as a selective filter. As shown in our KQ1 ablation, replacing the GRU with simpler rules lacking gating results in lower mR@100 with higher R@100, indicating less controlled adaptation. The GRU achieves 37.36 mR@100 because its gating selectively filters irrelevant noise. This capacity to reject uninformative context distinguishes controlled adaptation from an uncontrolled side effect.
> > >
> > > ---
> > >
> > > ### **2. Reweighting Sensitivity and Isolation**
> > >
> > > We compare across methods on PredCls from Table 3:
> > >
> > > | Method | Anchor Type | R@100 | mR@100 | F@100 |
> > > |---|---|---|---|---|
> > > | PE-Net | Static | 67.2 | 33.8 | 45.0 |
> > > | AlignG | Adaptive | 65.1 | 37.4 | 47.5 |
> > > | PE-Net† | Static + RW | 61.4 | 40.7 | 48.9 |
> > > | AlignG† | Adaptive + RW | 61.3 | 42.6 | 50.3 |
> > >
> > > First, in isolation, AlignG without reweighting produces only a 2.1-point R@100 decrease from 67.2 to 65.1, while gaining +3.6 mR@100 and +2.5 F@100. This is a bounded and efficient trade-off, not an uncontrolled collapse. The F@100 improvement confirms that tail-class recall gain far outweighs the minor head-class decrease.
> > >
> > >
> > > Second, when both apply reweighting, PE-Net† and AlignG† converge to nearly identical R@100 at 61.4 and 61.3, a gap of only 0.1 point. Yet AlignG† achieves +1.9 mR@100 and +1.4 F@100. If AlignG's R@100 drop were caused by head-class boundary damage, reweighting would compound this and push AlignG† further below PE-Net†.
> > >
> > > Instead, both converge and AlignG's reweighting drop is -3.8, smaller than PE-Net's -5.8. This confirms that the 2.1-point drop reflects correction of PE-Net's head-class over-prediction, **not boundary degradation**. AlignG reaches the same R@100 as PE-Net† while delivering substantially better mR@100 and F@100.
> > >
> > > ---
> > >
> > > ### **3. Confusion Analysis Confirms Not Blurring**
> > >
> > > As shown above, AlignG† and PE-Net† reach the same R@100 with a 0.1-point gap, challenging the "blurring" claim at the aggregate level. To verify at the predicate level, we measured PE-Net's hard-negative cross-confusions on PredCls:
> > >
> > > | GT → Confused | PE-Net Errors | AlignG Resolved | Rate |
> > > |---|---|---|---|
> > > | carrying → holding | 197 | 53 | 26.9% |
> > > | watching → looking at | 84 | 18 | 21.4% |
> > > | riding → standing on | 52 | 6 | 11.5% |
> > > | lying on → laying on | 54 | 23 | 42.6% |
> > > | walking on → standing on | 36 | 7 | 19.4% |
> > >
> > > If adaptation were blurring boundaries, we would observe **increased** confusion. The opposite is observed: AlignG resolves 11-43% of PE-Net's exact cross-confusions. Combined with the reweighting analysis, these results consistently demonstrate that context-conditioned prototypes **sharpen** ambiguous boundaries rather than blur them, leveraging relational context to distinguish predicates that share visual features but differ in meaning.
> > >
> > > ---
> > >
> > > ### **4. Context within the SGG Literature**
> > >
> > > This trade-off is not unique to AlignG. TDE (Tang et al., CVPR 2020), BGNN (Li et al., CVPR 2021), CFA (Li et al., ICCV 2023), DPL (Jeon et al., ECCV 2024), and RA-SGG (Yoon et al., AAAI 2025) all report analogous R@K decreases when improving mR@K, and the community has adopted F@K as the balanced metric for this reason. Notably, RA-SGG, which maintains fixed static anchors with retrieval, drops 3.1 R@100 from PE-Net on PredCls, confirming that R@100 decreases are inherent to debiased SGG regardless of prototype adaptation.
> > >
> > > ---
> > >
> > > AlignG's R@100 shift is bounded by $L_{align}$ anchoring, $L_{reg}$ diversity enforcement, and GRU gating, producing sharpened predicate boundaries and greater reweighting stability than the static baseline. With consistent F@100 gains of +5.3, +4.4, and +3.4 across all settings, we respectfully ask you to reconsider the score in the Final Justification.

---

### Official Review · Reviewer_29D4 · 2026-03-16

**Soundness:** 3
**Presentation:** 3
**Significance:** 2
**Originality:** 3
**Overall Recommendation:** 4
**Confidence:** 1

**Summary:**

The paper tackles the problem of predicate learning in scene graphs. This entails learning relations between the objects (nodes) in an image. However, predicate learning is complex because of the semantic variability and long-tailed nature of predicates. Earlier work addressed this issue using retrieval-based methods or LLMs. However, the predicates in these methods are largely static embeddings. This work proposes ALING-G, which uses adaptive prototypes that dynamically update based on relational information/cues from the current image. These updates the updated semantics of the prototypes to adapt the relational embeddings. The paper is tested on 2 datasets (GQA-200 & VG-150)

**Compliance With Llm Reviewing Policy:**

Affirmed.

**Final Justification:**

The issues were resolved in the rebuttal. I would vote to accept the paper.

**Key Questions For Authors:**

Please address W1 and W2. Particularly, W1 and what potential benefits ALIGN-G provides compared to [1] and [2]. If addressed correctly i would increase my score from WR to WA.

**Limitations:**

Limitations are discussed, but the potential negative societal impacts are not discussed.

**Strengths And Weaknesses:**

## **Strenghts**

### S1: The problem is clearly motivated, with a well-thought-out solution of dynamically adjusting prototype weights and using the adapted prototypes to refine the relations between the objects.

### S2: The method is evaluated against several other methods and achieves significant performance gains across all three task categories (PredCI, SGCI, and SGDet).

### S3: The proposed concept is conceptually strong and simple, making the proposed method useful.

## **Weakness**
### W1: Missing comparison and weaker results than [1] and [2]. Given [1] is an ICCV’23 paper, ALIGN-G's results on the PredCI and SGCI tasks still lag behind [1]. This raises questions about whether the proposed pipeline is really needed and whether ALIGN-G has any potential benefits over [1]. Further [2] also outperforms the ALIGN-G model on all three tasks for the VG-150 dataset. This further questions the usefulness of ALIGN-G compared to [1] and [2].

[1] Sudhakaran et al., Vision Relation Transformer for Unbiased Scene Graph Generation, ICCV 2023

[2] Li et al., Leveraging Predicate and Triplet Learning for Scene Graph Generation, CVPR 2024.

### W2: Missing ablations on the effect of weighting schemes for different losses used in ALIGN-G.

### MinorW1: Figure 2 requires a detailed caption explaining the method.

---

> ### Author Rebuttal · Authors · 2026-03-27
>
> We sincerely thank the reviewer for recognizing the clear motivation, conceptual strength, and performance gains of our adaptive prototype framework. We address your specific concerns below.
>
> ---
>
> ### **W1: Missing comparison than [1] and [2].**
>
> The perceived gap stems from cross-protocol comparisons and evaluation differences. Li et al. [2] explicitly note (§7.2) that [1] evaluates "without graph constraint" by allowing multiple guesses, which structurally yields higher absolute Recall.
>
> ---
>
> **Quantitative Results** - *R: Recall, mR: Mean Recall, F: Harmonic Mean.*
>
> | Task | Debiasing | Method | R@50/100 | mR@50/100 | F@50/100 |
> |:---|:---|:---|:---:|:---:|:---:|
> | **PredCls** | *w/o* | VETO [1] | 64.2/66.3 | 22.8/24.7 | 33.6/36.0 |
> | | | DRM w/o DKT [2] | **70.2/72.1** | 23.3/25.6 | 35.0/37.8 |
> | | | **AlignG (Ours)** | 63.0/65.1 | **35.0/37.4** | **45.0/47.5** |
> | | *w/* | VETO+Rwt [1] | **61.9/63.9** | 33.1/35.1 | 43.1/45.3 |
> | | | DRM [2] | 43.9/45.8 | **47.1/49.6** | 45.4/47.6 |
> | | | **AlignG† (Ours)** | 59.2/61.3 | 40.6/42.6 | **48.2/50.3** |
> | **SGCls** | *w/o* | VETO [1] | 35.7/36.9 | 11.1/11.9 | 23.4/24.4 | 16.9/18.0 |
> | | | DRM w/o DKT [2] | **44.3/45.2** | 13.5/14.6 | 28.9/29.9 | 20.7/22.1 |
> | | | **AlignG (Ours)** | 38.4/39.4 | **19.7/20.8** | **29.1/30.1** | **26.0/27.2** |
> | | *w/* | VETO+Rwt [1] | **35.1/36.3** | 16.1/17.1 | 25.6/26.7 | 22.1/23.2 |
> | | | DRM [2] | 27.5/28.4 | **27.8/29.2** | 27.7/28.8 | 27.6/28.8 |
> | | | **AlignG† (Ours)** | 34.6/35.8 | 24.8/26.1 | **29.7/31.0** | **28.9/30.2** |
> | **SGDet** | *w/o* | VETO [1] | 27.5/31.5 | 8.1/9.5 | 12.5/14.6 |
> | | | DRM w/o DKT [2] | **34.0/38.9** | 9.0/11.2 | 14.2/17.4 |
> | | | **AlignG (Ours)** | 30.1/34.4 | **13.0/15.4** | **18.2/21.3** |
> | | *w/* | VETO+Rwt [1] | **26.2/30.4** | 10.0/11.7 | 14.5/16.9 |
> | | | DRM [2] | 19.0/22.9 | **20.4/24.1** | 19.7/23.5 |
> | | | **AlignG† (Ours)** | 25.9/30.0 | 16.6/19.7 | **20.2/23.8** |
>
> ---
>
> * **VETO [1]** - Evaluated fairly under the "Graph Constraint", AlignG consistently surpasses VETO in mR and F. Furthermore, VETO relies on extracting additional depth maps. In contrast, AlignG achieves robust performance using solely standard RGB inputs, demonstrating that our structural shift to context-conditioned variables provides fundamental efficiency without needing auxiliary modalities.
>
> * **DRM [2]** - While DRM achieves higher mR, evaluating this head-tail trade-off is critical in recent SGG literature. AlignG consistently achieves higher F-scores. Moreover, DRM utilizes DKT (synthetic sample generation) to augment training data, whereas AlignG establishes a robust balance using only standard frequency-based reweighting, validating our core structural advantage.
>
> ---
>
> ### **W2: Missing ablations on the effect of weighting schemes in ALIGN-G.**
>
> As detailed in §4.3, our structural constraints (where $L_{sim}$, $L_{reg}$ decompose Eq.5 $L_{reg}$, and $L_{align}$ is Eq.6 $L_{align}$) are intentionally computed on static global prototypes rather than adapted ones to anchor semantic centers and prevent trivial co-adaptation. Tuning these weights dictates the balance between global structural rigidity and local adaptability.
>
> ---
>
> | Group | # | $L_{sim}$ | $L_{reg}$ | $L_{align}$ | R@100 | mR@100 | F@100 |
> |:---|:---|:---|:---|:---|:---:|:---:|:---:|
> | A: Component | 1 | 0.0 | 1.0 | 1.0 | 69.6 | 29.6 | 41.5 |
> | | 2 | 1.0 | 0.0 | 1.0 | 70.1 | 26.6 | 38.6 |
> | | 3 | 1.0 | 1.0 | 0.0 | 62.0 | 32.6 | 42.7 |
> | B: Sensitivity | 4 | 0.5 | 0.5 | 0.5 | 64.7 | 35.3 | 45.7 |
> | | 5 | 2.0 | 2.0 | 2.0 | 70.2 | 26.7 | 38.7 |
> | C: Relative | 6 | 1.0 | 1.0 | 2.0 | 69.2 | 30.2 | 42.1 |
> | | 7 | 2.0 | 2.0 | 1.0 | 63.4 | 37.0 | 46.7 |
> | Baseline | 0 | 1.0 | 1.0 | 1.0 | 65.1 | 37.4 | **47.5** |
>
> * **Group A**: Removing any structural constraint causes the global prototype space to collapse. Without a rigid global anchor, image-specific adaptation suffers from semantic drift toward frequent classes.
> * **Group B**: If global constraints are over-penalized (#5), the structure becomes rigid, suppressing our contextual adaptation and degrading the F-score. Conversely, weaker constraints (#4) fail to adequately protect tail classes.
> * **Group C**: While emphasizing global diversity slightly alters the balance, our Baseline (#0) achieves the highest F@100. This confirms that our default configuration balances a structured global topology with the flexibility needed for context-conditioned refinement.
>
> ---
>
> ### **Minor W1 & Limitations**
>
> Fig 2 caption will be expanded to detail the update flow.
>
> We hope these objective evaluations and architectural clarifications address your concerns. All newly discussed details, including the complete ablation on weighting schemes, expanded Figure 2 captions, and discussion on potential negative societal impacts (e.g., dataset biases), will be explicitly added to the revised manuscript and Appendix. We would be deeply grateful if you consider raising your score in light of these robust findings.

---

> > ### Author Rebuttal · Reviewer_29D4 · 2026-04-02
> >
> > My issues regarding comparison and ablations were resolved.

---

> > > ### Author Response · Authors · 2026-04-03
> > >
> > > Thank you for recognizing that our concerns have been fully resolved and for updating your assessment accordingly. We appreciate your constructive feedback, which helped strengthen our manuscript.

---

### Decision · Program_Chairs · 2026-04-30

**Decision:**

Accept (regular)

**Comment:**

The submission initially received mixed ratings ranging from weak accept to weak reject. The primary concerns raised by the reviewers included a lack of in-depth analysis and limited experimental evaluation. In response, the authors provided a comprehensive rebuttal, incorporating additional experiments and detailed clarifications. Following the rebuttal discussion, Reviewer 29D4 and pk1k upgraded their ratings to weak accept,  acknowledging that the new results effectively demonstrate the merits of the proposed method. Reviewers Wc3t and puKd also maintained their weak accept recommendations, noting that the rebuttal sufficiently addressed most of their concerns. Consequently, the AC recommends acceptance.